# PROTOKV: LONG-CONTEXT KNOWLEDGES ARE ALREADY WELL-ORGANIZED BEFORE YOUR QUERY

**Zhiyuan Yu[1]\***, **Shijian Xiao[1]\***, **Zhangyue Yin[2]**, **Xiaoran Liu[2]**, **Lekai Xing[1]**,
**Wenzhong Li[1]†**, **Nguyen Cam-Tu[1]**, **Sanglu Lu[1]**
[1] State Key Laboratory for Novel Software Technology, Nanjing University,
[2] School of Computer Science, Fudan University
`zhiyuan_yu@smail.nju.edu.cn, lwz@nju.edu.cn`

## ABSTRACT

Modern Large Language Models (LLMs) face fundamental challenges in processing long text sequences due to the quadratic complexity of attention mechanisms. Key-Value (KV) cache retention strategies mitigate this issue by selectively preserving salient KV pairs for autoregressive generation. However, existing methods fail to adequately and efficiently preserve the semantic integrity of the compressed representations. In this paper, we discover a prevalent phenomenon in LLM: within the key embedding space, while most tokens exhibit similarity with their contextual neighbors (we term position-determined tokens), a small subset of special tokens serving as semantic anchors consistently show local deviation property and form one or several clusters (we term semantic-anchored tokens). Motivated by this observation, we propose ProtoKV that separately processes these two token categories for KV cache compression. Within this framework, we first construct semantic prototypes based on the inherent characteristics of the two token categories, and subsequently form clusters of semantically similar tokens as basic compression units. This approach preserves semantic integrity with high computational efficiency. Experiments on LongBench demonstrate that ProtoKV achieves 2.11% higher accuracy than state-of-the-art methods under matched memory constraints. Our code can be available in https://github.com/yyy0959/ProtoKV.

## 1 INTRODUCTION

Large language models (LLMs) have become revolutionary in modern artificial intelligence (Brown et al., 2020; Chowdhery et al., 2023; Touvron et al., 2023), showcasing remarkable capabilities in dialogue (Li et al., 2024a), question answering (Ho et al., 2020) and reasoning (Wei et al., 2022). However, deploying LLMs under fixed-memory hardware presents major computational hurdles due to the Key-Value (KV) cache, which stores historical KV vectors to avoid recomputation but consumes memory scaling as $\mathcal{O}(b \cdot n)$, with respect to batch size $b$ and sequence length $n$.

To address this challenge, strategies have been proposed to optimize the KV cache. Architecture-level optimizations like MQA (Shazeer, 2019) and GQA (Ainslie et al., 2023) reduce memory via parameter sharing, and quantization approaches (Hooper et al., 2024b; Shao et al., 2024) lowers numerical precision. Both methods demonstrate limited efficacy when scaled to extremely long contexts. To this end, eviction strategies emerge to optimize the KV cache by prioritizing and retaining salient KV pairs in each head for generation. Importance of KV pairs is typically determined by various configurable schemes, including prior knowledge (e.g., "attention sink" in (Xiao et al., 2024)) and quantitative indicators like accumulative attention scores (Zhang et al., 2023). Considering that measuring token importance isolately may compromise semantic integrity, recent methods (Li et al., 2024b; Razzhigaev et al., 2025) retain both high-value tokens along with their surrounding context to better maintain semantic coherence, leading to a significant boost in KV cache retention efficiency.

---

\* Equal contribution
† Corresponding author

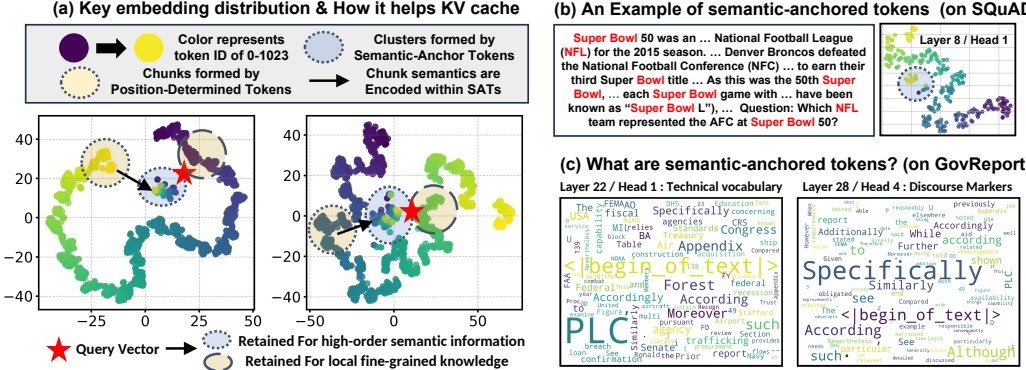

Figure 1: **(a)** T-SNE visualization of key embedding. As illustrated, semantic-anchored tokens typically form clusters within the key embedding space. **(b)** We used a sequence from SQuAD and highlighted the tokens corresponding to semantic anchors in a specific attention head, which are marked in red. **(c)** Different attention heads are dedicated to distinct semantic-anchored tokens.

However in natural language, proximity in position does not imply semantic similarity. Existing work has observed that when LLMs process input sequences, certain sequential information will be compressed into anchor tokens, e.g., label words for in-context learning (Wang et al., 2023) or some punctuation marks (Razzhigaev et al., 2025). These tokens share similar properties and play analogous roles during inference, suggesting that they should be treated as a semantically cohesive unit (either collectively discarded or retained). In practice, however, the diversity of anchor tokens poses a significant challenge for systematically identifying and categorizing them.

Fortunately, we find that **LLMs have inherently identified and organized these anchor tokens during prefilling stage**. Specifically, by analyzing the key embedding distributions, we identify that tokens in long-context prompts mainly fall into two distinct categories. **(i) Position-Determined Tokens (PDT)**: These tokens exhibit strong key similarity with their contextual neighbors, corresponding to the majority tokens where proximity implies semantic relatedness; and **(ii) Semantic-Anchored Tokens (SAT)**: These tokens violate such local adherence and demonstrate **local deviation property**. Previous studies have primarily focused on PDTs, suggesting that rotary position encodings (Su et al., 2023) cause key embeddings to exhibit manifold characteristics (Zandieh et al., 2024; Liu et al., 2025b;c), while overlooking the existence of SATs. In this paper, we discover that **although SATs are few in number, they generally possess good clustering properties and serve as "semantic anchors" for long-context inference**. In Section 3, we conduct a detailed quantitative study to investigate and analyze such properties of SATs, while also explaining the reasons behind such distribution pattern of the key embeddings. In Figure 1, we illustrate the characteristics of this distribution through T-SNE visualization, accompanied by examples for SATs.

Through further exploration, we find it imperative for KV cache compression to simultaneously accommodate the aggregated information encapsulated by SATs and the fine-grained information inherent in PDTs. To this end, we propose ProtoKV, a novel KV cache retention strategy that first constructs hybrid semantic prototypes based on the inherent characteristics of the two token categories, and then dynamically reassigns importance scores. Specifically, we employ **outlier degree** metric to obtain a high-purity set of SATs, and apply locality-sensitive hashing to bucket them to form SAT semantic prototypes that encapsulate anchor-level semantic information. For PDTs, we partition them into contiguous chunks to form PDT semantic prototypes that are prioritized for fine-grained information retrieval. Finally, an observation window-based selection mechanism is adopted to retain the semantic clusters most relevant to subsequent queries. Through comprehensive comparisons with existing KV cache retention strategies, we demonstrate that our approach not only preserves the semantic integrity in a more reasonable manner, but also achieves superior robustness with higher computational efficiency. This dual advantage remains unattainable by existing methods.

Extensive experiments demonstrate that ProtoKV achieves excellent performance on both real-world and synthetic tasks. Specifically, ProtoKV delivers state-of-the-art accuracy (with an average gain of 2.11%) on LONGBENCH, and also outperforms other baselines on the RULER benchmark. In the Needle In A Haystack evaluation, ProtoKV maintains 97.3% retrieval accuracy with only 1.6%

KV cache retention, demonstrating its powerful information retrieval capability. Moreover, ProtoKV shows complementary benefits when combined with budget allocation methods.

## 2 PRELIMINARY

### 2.1 PROBLEM FORMULATION

Consider an autoregressive LLM with $L$ layers and $H$ attention heads. Let $\mathbf{x}_t \in \mathbb{R}^d$ denote the input token embedding at decoding step $t$. To reduce recomputation, key-value pairs from previous steps are stored in a KV-Cache for each of the $H$ heads. The attention output $\mathbf{o}_t^{(h)} \in \mathbb{R}^{d_h}$ at step $t$ is computed via a softmax attention mechanism using the current query vector and the cached key and value matrices. For each head $h$, our objective is to find compressed representations $\{\tilde{\mathbf{K}}_{1:t}^{(h)}, \tilde{\mathbf{V}}_{1:t}^{(h)}\}_{h=1}^{H}$ under a budget size $\mathcal{B}$, such that $\|\tilde{\mathbf{K}}_{1:t}^{(h)}\|_0 = \|\tilde{\mathbf{V}}_{1:t}^{(h)}\|_0 = \mathcal{B}$, and the approximation error $\|\mathbf{o}_t^{(h)} - \tilde{\mathbf{o}}_t^{(h)}\|_2$ is minimized. Here $\tilde{\mathbf{o}}_t^{(h)}$ denotes the approximate output using the compressed KV pairs.

### 2.2 KV CACHE RETENTION

We focus on compressing long text prompts during the prefilling stage, i.e., the compression of $\{\mathbf{K}_{1:N}^{(h)}, \mathbf{V}_{1:N}^{(h)}\}_{h=1}^{H}$. Theoretically, the optimal strategy for KV cache retention should prioritize tokens that consistently contribute to the model's attention distribution throughout the entire generation process, which can be measured through *inference-stage cumulative attention* defined as follows:

**Definition 1** (Inference-stage Cumulative Attention). *Given attention head $h$, let $T$ denotes the total number of generation steps and $\mathbf{q}_t^{(h)}$ the queries in step $t$. $\{\mathbf{k}_i^{(h)}\}_{i=1}^{N}$ denote the key vectors of all $N$ context tokens. The inference-stage cumulative attention $\mathcal{A}_i^{(h)}$ for context token $i$ is defined as:*

$$\mathcal{A}_i^{(h)} = \sum_{t=1}^{T} softmax\left(\frac{\mathbf{q}_t^{(h)\top} \mathbf{k}_i^{(h)}}{\sqrt{d_h}}\right) \tag{1}$$

Precomputing $\mathcal{A}_i^{(h)}$ is infeasible since $\mathbf{q}_t^{(h)}$ is unpredictable during autoregressive generation. As a result, strategies using accumulative attention scores (Zhang et al., 2023; Chen et al., 2024; Liu et al., 2024a; Zeng et al., 2024) offer more flexibility. However, most of their methods measure token importance isolatedly, which may compromise semantic integrity for compressed KV pairs.

### 2.3 SEMANTIC-LEVEL COMPRESSION

The semantic-level compression paradigm addresses the limitations of token-level compression by treating coherent semantic units as atomic elements in KV cache management. Let $\mathcal{C} = \{C_1, ..., C_k\}$ denote a partition of the KV sequence into $k$ semantic clusters, where each cluster $C_i$ contains contiguous or semantically related tokens. The compression objective is to preserve or discard entire clusters to maintain contextual integrity. Typically, clusters are ranked by importance scores $\psi^{(h)}(C_i)$. The top-$k'$ clusters ($k' \le k$) are retained to satisfy budget size $\mathcal{B}$:

$$\sum_{i=1}^{k} \mathbb{I}[\psi^{(h)}(C_i) \ge \psi_{\text{rank}=k'}^{(h)}] \cdot |C_i| \le \mathcal{B}. \tag{2}$$

The compressed KV pairs $\{\tilde{\mathbf{K}}_{1:N}^{(h)}, \tilde{\mathbf{V}}_{1:N}^{(h)}\}$ retain only tokens from selected clusters, with a binary mask $\mathbf{M}^{(h)} \in \{0,1\}^t$ indicating preservation ($\mathbf{M}_i^{(h)} = 1$ if $i$ belong to one of the selected clusters):

$$\tilde{\mathbf{K}}_{1:N}^{(h)} = \mathbf{K}_{1:N}^{(h)} \odot \mathbf{M}^{(h)}, \quad \tilde{\mathbf{V}}_{1:N}^{(h)} = \mathbf{V}_{1:N}^{(h)} \odot \mathbf{M}^{(h)}. \tag{3}$$

For instance, SnapKV (Li et al., 2024b) and ChunkKV (Liu et al., 2025a) group tokens into contiguous chunks, and SentenceKV (Zhu et al., 2025) uses sentence boundaries for $C_i$. However in natural language, proximity in position does not necessarily imply semantic similarity. Clustering-based KV compression approaches (Liu et al., 2024b; Hooper et al., 2024a) employ K-means clustering on key embeddings, but are plagued by inefficiency and poor robustness. By contrast, our proposed ProtoKV is capable of simultaneously preserving semantic integrity and maintaining high efficiency.

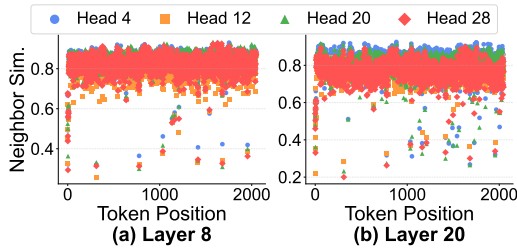
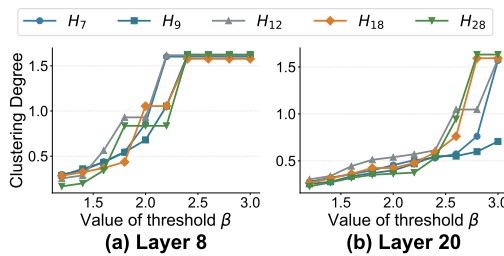

Figure 3: $\kappa$-Neighborhood Similarity tend to attain low values for a small subset of input tokens. Detailed results and analysis are in Appendix B.

Figure 4: SATs form progressively compact clustering with the increasing the threshold $\beta$. Detailed results and analysis are in Appendix C.

## 3 LOCAL DEVIATION PROPERTY ANALYSIS

### 3.1 PROPERTY ANALYSIS FOR SEMANTIC-ANCHOR TOKENS

In this section, we highlight a prevalent phenomenon during the prefilling stage of LLMs: **within the key embedding space, the vast majority of tokens exhibit high similarity to their local neighbors, while a compact subset demonstrably violates this locality prior**, which we term **Local Deviation Property**. Crucially, we find these spatially anomalous tokens consistently form one or several clusters, and play a pivotal role in long-context generation as **semantic anchors**. To identify these tokens, we first define the neighborhood similarity and outlier degree as follows.

**Definition 2** ($\kappa$-Neighborhood Similarity & Outlier Degree). *For token $i$ with its key embedding $\mathbf{k}_i^{(h)}$ in attention head $h$, we define its $\kappa$-neighborhood similarity as follows:*

$$\mathcal{S}_\kappa^{(h)}(i) = \frac{1}{2\kappa+1} \sum_{j=i-\kappa}^{i+\kappa} \cos(\mathbf{k}_i^{(h)}, \mathbf{k}_j^{(h)}), \tag{4}$$

*In subsequent experiments, unless otherwise specified, we set $\kappa = 5$ by default. To identify tokens with local heterogeneity in a normalized manner, we define outlier degree for token $i$ as:*

$$\Theta^{(h)}(i) = \left(\mathcal{S}_\kappa^{(h)}(i) - \mathbb{E}[\mathcal{S}_\kappa^{(h)}(i)]\right)/\sqrt{\mathbb{V}[\mathcal{S}_\kappa^{(h)}(i)]}, \tag{5}$$

*with $\mathbb{E}$ and $\mathbb{V}$ represent the mean and variance of the neighborhood similarity across all input tokens.*

Figure 3 reveals that although most tokens maintain high neighborhood similarity, there always exists a small subset exhibiting markedly lower values. We term tokens violating the locality property as **Semantic-Anchored Tokens (SATs)** due to its aggregation of semantic information from partial sequences, while tokens conforming to locality are termed **Position-Determined Tokens (PDTs)**. It is important to note that the boundary between SATs and PDTs is often ambiguous in practice (as illustrated in Figure 2); therefore, our analysis focuses primarily on the evolutionary trends of token properties along a continuum of locality adherence/deviation. We find that the following findings remain valid under the framework of Multi-Query Attention, which is verified in Appendices B to E.

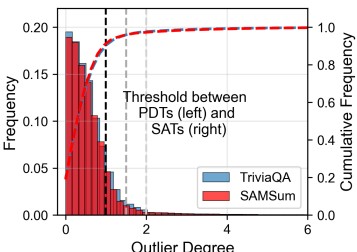

Figure 2: Outlier Degree distribution.

**Property 1: SATs are *clustered*.** To quantitatively validate the clustering property of SATs, by setting a threshold $\beta$, we group tokens with $\Theta(i) > \beta$ into cluster $\mathcal{C}$ as SATs. Obviously, a larger $\beta$ indicates higher purity of the selected SATs. We then define the clustering degree of $\mathcal{C}$ as follows:

**Definition 3** (Clustering Degree). *Given a cluster $\mathcal{C}$ of tokens with their key representations $\{\mathbf{k}_i^{(h)}\}$, we define the intra-cluster similarity $\mathcal{S}_{intra}(\mathcal{C})$ and inter-cluster similarity $\mathcal{S}_{inter}(\mathcal{C})$ as:*

$$\mathcal{S}_{intra}(\mathcal{C}) = \frac{1}{|\mathcal{C}|^2} \sum_{i,j \in \mathcal{C}} \cos(\mathbf{k}_i^{(h)}, \mathbf{k}_j^{(h)}), \quad \mathcal{S}_{inter}(\mathcal{C}) = \frac{\sum_{i \in \mathcal{C}, j \notin \mathcal{C}} \cos(\mathbf{k}_i^{(h)}, \mathbf{k}_j^{(h)})}{|\mathcal{C}|(N - |\mathcal{C}|)}, \tag{6}$$

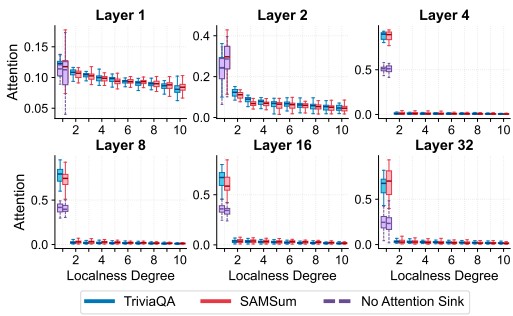
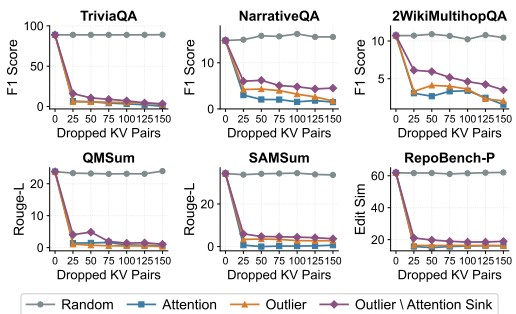

Figure 5: Attention heads within all layers focus on tokens with higher outlier degree. Detailed results and analysis are in Appendix E.

Figure 6: Dropping KV pairs with highest outlier degrees can cause sharp performance drop. Other results and analysis are in Appendix E.

*where $N$ denotes the total number of tokens. The clustering degree of $\mathcal{C}$ that measures its compactness is then defined as $\Gamma(\mathcal{C}) = \mathcal{S}_{intra}(\mathcal{C})/\mathcal{S}_{inter}(\mathcal{C})$.*

Building upon these formulations, we systematically evaluate the clustering properties of SATs selected under varying $\beta$ thresholds. Figure 4 shows the results on *TriviaQA* for layers (8 and 20) in `Llama2-7B-chat`, where increasing $\beta$ leads to a more compact cluster of $\mathcal{C}$ with larger $\Gamma(\mathcal{C})$. Notably, when $\beta$ exceeds a critical threshold, the clustering degree exhibits a sharp increase. This suggests that in the key embedding space, tokens with high outlier degrees tend to cluster. Moreover, we observe that in rare cases (e.g., when long-text semantics exhibit significant discontinuity), these SATs may form multiple distinct clusters, with representative examples provided in Appendix D.

**Property 2: SATs are *salient* for generation.** To validate SATs' critical role in long-context inference, we analyze two complementary aspects. First, we examine if SATs are retrieved more frequently during long-text generation. We stratify tokens into 10 groups by their outlier degree $\Theta(i)$ (Eq.5) and compute summed inference-stage cumulative attention (Eq.1) per group for token significance measurement. As Figure 5 shows, the top 10% outlier tokens account for over 60% of cumulative attention in deeper layers, indicating that SATs act as persistent **anchors of attention**.

On the other hand, we study the impact of removing these SATs. Figure 6 shows that pruning key-value pairs corresponding to highest outlier degree $\Theta^{(h)}(i)$ per head significantly degrades LLM's performance for long-context inference, which is similar to pruning by inference-stage cumulative attention. Random removal has no measurable effect. This confirms SATs' functional equivalence to the model's inherent attention prioritization.

**Conclusion.** Previous studies have focused mainly on PDTs (Zandieh et al., 2024; Liu et al., 2025b;c). In this paper, we **first** detected the presence of outliers within them, which spontaneously deviate from their surrounding neighbors and cluster together, serving as anchors of attention across different tasks. Considering their properties, we name them "**semantic anchors**".

## 3.2 FURTHER EXPLORATION

**How SATs Cluster.** In real-world experiments, we observe that SATs predominantly form a single, tightly-clustered group in the key embedding space across most attention heads. However, the clustering behavior exhibits certain variations under specific conditions: in the initial layers of the model, SATs may not yet emerge distinctly, resulting in a near-continuous distribution of key embeddings without clear outliers. In some cases, SATs are clustered but exhibit a loosely-organized structure; and in low-coherence texts such as multi-document data, SATs occasionally form multiple well-separated semantic groups. Despite these variations, our subsequent experiments in Section 5.3 demonstrate that assuming SATs form a single compact cluster remains highly effective for KV cache compression. Comprehensive analysis for each scenario is provided in Appendix D.

**Causal Mechanisms Analysis.** We attribute the above phenomenon to the following reason. First, modern LLMs commonly employ Rotary Position Embedding (RoPE) (Su et al., 2023) to inject positional information into keys, which leads to local similarity in the key representations of adjacent tokens (Wang et al., 2024b; Liu et al., 2025c).During pre-training, different attention heads

spontaneously specialize in capturing distinct linguistic patterns (Conmy et al., 2023; Syed et al., 2023), and thus attend to their corresponding salient tokens. To efficiently access relevant value embeddings, LLMs learn to cluster these salient keys in the key space. Such clustering mechanism enables the heads to simultaneously assign higher attention weights to semantically relevant tokens during generation. Meanwhile, tokens that are less relevant to a head's specific role remain locally constrained due to the inductive bias introduced by RoPE.

**"Semantic Anchors" vs "Attention Sink"** (Xiao et al., 2024) observed the "attention sink" phenomenon where the initial tokens disproportionately absorb global semantic information. In Appendix B, we note that attention sinks generally exhibit high outlier degrees. However, our concept of semantic anchors is broader in scope. As shown in Figures 5 and 6, even after excluding attention sinks, the remaining SATs continue to play an indispensable role in long-context generation.

**"Semantic Anchors" vs "Label words in In-context Learning"** (Wang et al., 2023) demonstrated that in in-context learning scenarios, label words act as anchors that absorb sample semantics for text classification tasks. Our study reveals that these label word tokens also exhibit significant outlier characteristics. Detailed experimental results are presented in the Appendix F.

# 4 METHODOLOGY

## 4.1 MOTIVATION

Considering the importance of SATs for long-context tasks, an intuitive approach is to retain KV pairs based on their outlier degree in descending order (we term SATKV). Compared to prior approaches, SATKV offers the advantage of eliminating the reliance on attention matrix, thereby reducing compu-

Table 1: Performance for Llama3-8B (256 budget).

| Baseline | Dataset | | | |
| --- | --- | --- | --- | --- |
| | NrtvQA | HotpotQA | SAMSum | Lcc |
| SLM | 17.98 | 37.83 | 34.82 | 54.84 |
| H2O | **23.67** | 41.57 | **40.19** | 57.52 |
| SnapKV | 23.32 | **42.70** | 39.78 | **60.27** |
| SATKV | 21.12 | 40.84 | 38.06 | 58.46 |

tational overhead (see Appendix H). However, we find this approach underperforms baselines like SnapKV (Table 1). We attribute this to two factors: First, it neglects PDT selection patterns. Figure 5 shows PDTs comprise around 40% of attention, which necessitates simultaneous optimization for PDTs retention especially when the KV budget exceeds SAT capacity. Second, similar to accumulative attention methods like H2O, it assesses tokens individually and fails to maintain semantic coherence. Thus, we require a retention strategy that preserves semantic coherence while integrating both high-order semantic information (SATs) and local fine-grained knowledge (PDTs).

To this end, we introduce ProtoKV, a novel method that constructs semantic prototypes by leveraging the inherent characteristics of both token categories. These prototypes form clusters of semantically similar tokens, serving as fundamental compression units. We compare our method with existing chunk-based and cluster-based approaches, demonstrating that our solution simultaneously preserves semantic integrity while improving efficiency. Pseudo-code for ProtoKV is provided in Appendix I.

## 4.2 HYBRID SEMANTIC PROTOTYPE CONSTRUCTION

Considering that drawing a hard distinction between PDTs and SATs is often challenging, instead of categorizing individual tokens, we propose extracting semantic prototypes holistically for patterns capture. Given an input sequence of $n$ tokens with corresponding key vectors $\mathcal{T} = \{\mathbf{k}_t\}_{t=1}^n \subseteq \mathbb{R}^{d_k}$, we first calculate the outlier degree $\Theta(i)$ of each token $i$ according to Eq.(5). Tokens with top-$p$ $\Theta(i)$ are identified as candidate SATs $\mathcal{O} = \{\mathbf{k}_j\}_{j=1}^p$. To better handle loosely-cluster or multi-cluster scenarios, for these identified outlier tokens, we allocate them into $u$ hash buckets. Specifically, the key vector for $j$th token is projected into low-dimensional space using Random Fourier Features (RFF) mapping $\phi : \mathbb{R}^{d_k} \to \mathbb{R}^r$ with Gaussian kernel approximation:

$$\phi(\mathbf{k}_j) = \sqrt{\frac{2}{r}} \cos\left(\mathbf{W}\mathbf{k}_j + \mathbf{b}\right), \quad \text{with } \mathbf{W} \sim \mathcal{N}(0, \gamma^2 I) \text{ and } \mathbf{b} \sim \text{Uniform}(0, 2\pi) \quad (7)$$

The real-valued projections are then binarized to $\{0, 1\}$ codes and subsequently interpreted as an $r$-bit integer for bucket allocation:

$$\mathbf{h}_j = \mathbb{I}\left(\phi(\mathbf{k}_j) > 0\right) \in \{0, 1\}^r, \quad \mathcal{H}(\mathbf{k}_j) = \left(\sum_{i=1}^r 2^{r-i} h_j^{(i)}\right) \bmod u, \quad (8)$$

where $\mathbb{I}(\cdot)$ denotes the element-wise indicator function, $h_j^{(i)}$ the $i$th bit of $\mathbf{h}_j$, and $u$ the total number of hash buckets. To ensure clustering effectiveness, we typically require that $u = 2^r$. This binary-to-decimal conversion preserves the Hamming distance between original codes while enabling efficient bucket indexing. For other tokens of $\mathcal{T} \setminus \mathcal{O}$, we partition them into $v$ consecutive chunks $\{\mathcal{C}_m\}_{m=1}^v$ of equal length $\lfloor (n-p)/v \rfloor$. Based on this, we construct hybrid semantic prototypes of PDTs and SATs as follows:

$$c_m^{(\text{PDTs})} = \frac{\sum_{\mathbf{k}_t \in \mathcal{C}_m} \mathbf{k}_t}{\| \sum_{\mathbf{k}_t \in \mathcal{C}_m} \mathbf{k}_t \|_2}, \quad c_s^{(\text{SATs})} = \frac{\sum_{\mathbf{k}_j \in \mathcal{B}_s} \mathbf{k}_j}{\| \sum_{\mathbf{k}_j \in \mathcal{B}_s} \mathbf{k}_j \|_2} \tag{9}$$

The hybrid semantic prototypes $\mathcal{M}$ is then obtained via: $\mathcal{M} = \{c_m^{(\text{PDTs})}\}_{m=1}^v \cup \{c_s^{(\text{SATs})}\}_{s=1}^u$. In Appendix D, we show that our approach for constructing semantic prototypes effectively accommodates distinct patterns of key embedding distributions.

## 4.3 KV Cache Retention via Prototype-Guided Attention

$\mathcal{M}$ constructs semantic prototypes for both higher-level semantic anchors ($c^{(\text{SATs})}$) and fine-grained textual details ($c^{(\text{PDTs})}$). For each token $\mathbf{k}_t$, the pattern to which it belongs (denoted as $c(\mathbf{k}_t)$) is determined by the semantic prototype with the highest cosine similarity to it, formally:

$$c(\mathbf{k}_t) = \arg\max_{c \in \mathcal{M}} \frac{\mathbf{k}_t^\top c}{\|\mathbf{k}_t\|_2 \|c\|_2}. \tag{10}$$

This assignment machanism divides all tokens into $n$ clusters $\{C_j\}_{j=1}^{u+v}$, and we calculate the importance score $\psi^{(h)}(C_j)$ for each through an observation window-based selection mechanism (Li et al., 2024b). Given an input sequence with its length $L_{\text{prompt}} = L_{\text{prefix}} + L_{\text{obs}}$, where $L_{\text{obs}}$ denotes the observation window at the sequence end, for each attention head $h \in [H]$, $\psi^{(h)}(C_j)$ is defined as:

$$\psi^{(h)}(C_j) = \sum_{i \in C_j} \sum_{m=L_{\text{prefix}}+1}^{L_{\text{prompt}}} \mathbf{q}_m^\top \mathbf{k}_i. \tag{11}$$

The compressed KV representations are then constructed according to Section 2.3.

## 4.4 Comparison with Existing KV Cache Compression Strategies

In this section, we compare our ProtoKV with existing chunk-based and cluster-based, showcasing that it ensures more rational semantic coherence while maintaining compression efficiency.

**Chunk-based strategies** Chunk-based strategies assume that semantically similar tokens usually appear in contiguous sequences. As a result, they partition input text into chunks and perform uniform retention/eviction operations on tokens within each chunk. However, our discovery of SATs reveals limitations in this assumption. For instance, as noted in (Razzhigaev et al., 2025), certain punctuation marks carry critical information transmission and memory functions, so their semantically similar counterparts should be other functionally equivalent punctuation marks. However, chunk-based methods fail to preserve the semantic integrity of such tokens.

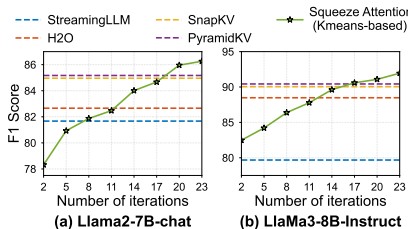

Figure 7: Clustering-based strategy requires 20 iterations before surpassing existing KV cache compression methods on *TriviaQA*.

**Cluster-based strategies** Clustering-based strategy (Liu et al., 2024b; Hooper et al., 2024a) global captures semantically similar tokens through clustering within the key embedding space. Although these methods can identify SATs, their iterative clustering process requires excessive computation. As illustrated in Figure 7, Squeeze Attention (Hooper et al., 2024a) takes over *20* iterations to outperform baselines (compress only during the prefilling stage). Moreover, prototype clustering typically demands high-quality initialization, as poor initialization can cause the iterative process to converge to a local optimum or lead to highly imbalanced token number distribution among clusters (Arthur & Vassilvitskii, 2007). By comparison, our ProtoKV directly obtains high-quality cluster centers by leveraging the properties of SATs and PDTs, thus reducing the computational complexity from $O(ent)$ to $O(nt)$ for $e$ iterations achieving $t$ clusters.

## 5 EXPERIMENT

### 5.1 IMPLEMENTATION

**Dataset** We primarily use *LongBench* (Bai et al., 2024) dataset to assess the performance of ProtoKV on tasks involving long-context inputs. LongBench comprises 14 English tasks and 2 code-related tasks, with an average length ranging from 5k to 15k tokens. We also use *Ruler* (Hsieh et al., 2024) as benchmark. A detailed description of datasets is provided in the Appendix J.

**Baseline** We benchmark our method against StreamingLLM (Xiao et al., 2024), H2O (Zhang et al., 2024d), SnapKV (Li et al., 2024b), PyramidKV (Cai. et al., 2024) and ChunkKV (Liu et al., 2025a). We use open-sourced LLMs include the Llama family (Llama-2-7B-chat, LlaMA-3-8B-instruct) and Mistral-7B-Instruct-v0.2, which can handle up to 32k context length. Detailed description of these three LLMs is provided in the Appendix K. In Appendix S, we also evaluate the performance of ProtoKV on two recent LLMs: Phi-3.5-mini-instruct and Mistral-7B-Instruct-v0.3.

**Experiment Setup** All experiments use two NVIDIA 3090 GPUs (48GB total) with consistent prompts across datasets. The Operating system is ubuntu version 20.04.2 with CPU AMD Ryzen Threadripper PRO 5945WX (12-Cores). KV cache budget for evaluation range from $64$ to $512$. Our configuration balances experimental uniformity with task-specific optimizations.

### 5.2 RESULT ANALYSIS

#### 5.2.1 REAL-WORLD AND SYNTHETIC BENCHMARKS

Figure 8 demonstrate the experimental results on *LongBench* across diverse KV cache configurations. Generally, our method maintains the best performance between 64-512 budgets, with an average improvement of 2.11%. As illustrated, ProtoKV outperforms Sota baselines by 0.35% to 4.27% across diverse budget sizes. Task-specific experimental results are reported in Appendix L. Additionally, Table 9 presents the performance of our ProtoKV on the Ruler dataset (NAIH and QA subsets), demonstrating that ProtoKV consistently achieves either the best or second-best outcomes. Due to its more rational partitioning of semantic clusters, our method achieves superior compression performance compared to both SnapKV and ChunkKV. In appendix P, we further report the performance comparison on RULER benchmark with a 16K/32K context length.

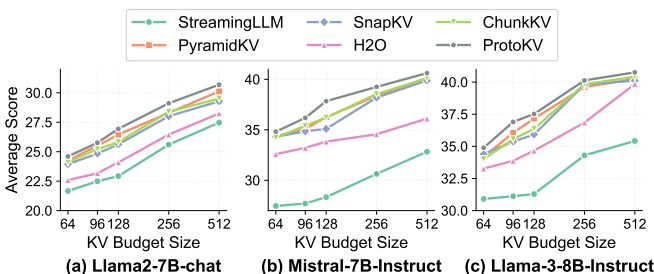

| Method | Dataset | |
|---|---|---|
| | **NIAH** | **QA** |
| FullKV | 99.6 | 46.0 |
| SLM | 3.6 / 17.4 | 7.6 / 10.2 |
| H2O | 30.4 / 57.2 | 38.0 / 42.0 |
| SnapKV | 97.2 / **99.6** | 42.0 / **43.8** |
| ChunkKV | 97.2 / **99.6** | 42.4 / 42.2 |
| PyramidKV | 97.6 / 99.4 | 41.2 / 43.4 |
| ProtoKV | **98.2** / **99.6** | **42.8** / 43.6 |

Figure 8: Experimental results on LongBench under different KV cache budget conditions. Average score is reported across 16 tasks.

Table 2: Performance (%) on Ruler for Llama3-8B with budget size 128/512.

We further compare ProtoKV with ClusterKV (Liu et al., 2024b), Squeezed-Attention (Hooper et al., 2024a), and SentenceKV (Zhu et al., 2025). These methods involve dynamically loading or offloading KV pairs during the inference stage. To ensure a fair comparison, we design two experimental configurations: one constrains these baseline methods to perform KV pair retention operations exclusively during the prefilling phase, while the other modifies ProtoKV to dynamically recall tokens at the granularity of semantic clusters during the decoding phase. As shown in Table 3, ProtoKV consistently outperforms other methods

| | Performance | |
|---|---|---|
| **Approach** | **Decoding** | **Prefilling** |
| SentenceKV | 40.82 | 40.21 |
| ClusterKV | 41.32 | 40.45 |
| SqueezeAtt | 41.47 | 40.42 |
| ProtoKV | **42.59** | **40.76** |

Table 3: Longbench performance for Llama3-8B under 512 budget.

under both configurations. This advantage stems from its more robust and higher-quality clustering results compared to K-means, which ultimately enhances its performance over existing cluster-based approaches.

### 5.2.2 NEEDLE IN A HAYSTACK

We conduct Needle In A Haystack experiment (Liu et al., 2024c; Fu et al., 2024a) using LlaMA-3-8B-Instruct with up to 8K context length. This task requires precise information retrieval from extensive contexts, simulating real-world scenarios where relevant data is buried among vast irrelevant information.We compare other KV Cache techniques at a consistent cache budget size of 128 (a retention ratio up to 1.6%). Results in Figure 9 indicate that StreamingLLM and H2O almost collapses on retrieval task. Our ProtoKV attains 97.3% accuracy, which outperforms SnapKV (94.2%). Interestingly, we observe that in the retrieval head (Wu et al., 2024), the needle text exhibits consistently higher outlier degrees than the haystack text (detailed results in Appendix G). This indicates that LLMs may detect semantic inconsistencies during the prefilling phase.



Figure 9: Results of the **Needle In A HayStack** experiment, where LLMs are required to retrieve a target sentence ("needle") **inserted** in long documents. The x-axis represents the context length while y-axis the depth where the needle is inserted. E.g., context length of 4000 and depth of 11.0 implies that the needle is inserted at location $4000 \times 11\% = 440$ in the sequence. The color indicates retrieval accuracy, the greener, the better.

### 5.3 FURTHER DISCUSSION

| Tasks | ProtoKV | ProtoKV + LA. | ProtoKV + HA. |
|---|---|---|---|
| SDQA | 32.87 | $\underline{33.12}_{\uparrow(0.54\%)}$ | $\mathbf{33.58}_{\uparrow(1.02\%)}$ |
| MDQA | 25.31 | $\underline{25.72}_{\uparrow(1.46\%)}$ | $\mathbf{26.14}_{\uparrow(1.24\%)}$ |
| SUM | 24.41 | $\underline{24.89}_{\uparrow(1.47\%)}$ | $\mathbf{25.29}_{\uparrow(1.15\%)}$ |
| Few shot | 66.43 | $\underline{66.85}_{\uparrow(1.21\%)}$ | $\mathbf{67.34}_{\uparrow(1.65\%)}$ |
| SYN | 37.15 | $\underline{37.43}_{\uparrow(0.82\%)}$ | $\mathbf{37.91}_{\uparrow(1.73\%)}$ |
| Code | 59.38 | $\underline{59.77}_{\uparrow(4.08\%)}$ | $\mathbf{60.16}_{\uparrow(3.27\%)}$ |

Table 4: **Compatibility Analysis** on Mistral-7B-Instruct with KV budget 256, $\uparrow(\cdot)$ denoting the improvement compared with SnapKV+*LA./HA.*.

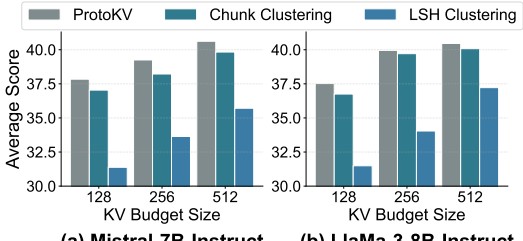

Figure 10: **Ablation Study** for different KV budgets.

**Compatibility Analysis** Methods for KV cache budget allocation (Wang et al., 2024a; Qin et al., 2025; Feng et al., 2024; 2025) intelligently distribute memory by the importance of each layer or head. We study their compatibility with our method, focusing on two strategies: *Layer-wise Allocation (LA.)* (Nawrot et al., 2024) and *Head-wise Allocation (HA.)* (Feng et al., 2024). As shown in Table 4, **ProtoKV** demonstrates strong compatibility, with $ProtoKV + LA./HA.$ achieving consistent improvement compared to $SnapKV + LA./HA.$.

**Ablation Study** We compare ProtoKV with its two variants for ablation study: *chunk-clustering* that only uses chunk-based aggregation to obtain semantic prototypes, and *LSH-clustering* that buckets and groups all tokens via LSH in Eq. 8. Two variants adopt the same number of semantic prototypes (i.e., cluster count) as the original ProtoKV, along with other basic settings. Figure 10 shows both variants reduce KV cache compression performance, especially LSH-clustering.

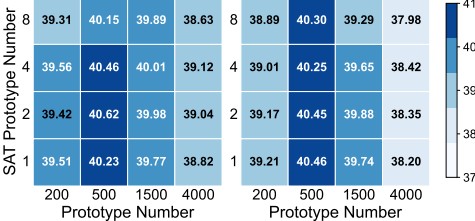

Figure 11: Average results on LongBench for Mistral-7B (LEFT) and Llama3-8B (RIGHT).

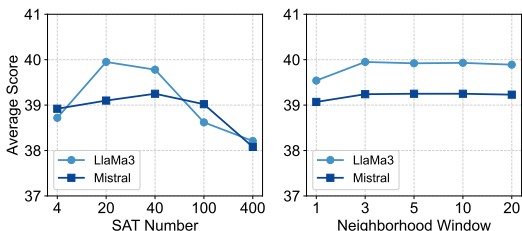

Figure 12: Average results on LongBench for Llama3-8B with KV budget size of 256.

**Hyperparameter Analysis.** As shown in Figure 11, optimal performance is achieved with around 500 prototypes, whereas SAT prototypes number require only 2–4. Figure 12 shows that selecting SATs via outlier degree requires merely 20–40 to optimize ProtoKV performance, since additional SAT prototypes may introduce PDT noise and impair clustering effectiveness. Moreover, the choice of neighborhood window size $\kappa$ in outlier degree computation shows negligible influence on the overall performance.

**Computation Cost Analysis** Figure 13 illustrates the computational cost of our ProtoKV. It can be observed that our time consumed in evaluating the importance of semantic clusters $\psi$ after prefilling is comparable with SnapKV, whereas cluster-based methods require up to $3.9 \times$ time overhead. Furthermore, the overall computational time of ProtoKV remains relatively stable across different prototype numbers, which is reported here as the average time over various tasks on LongBench.

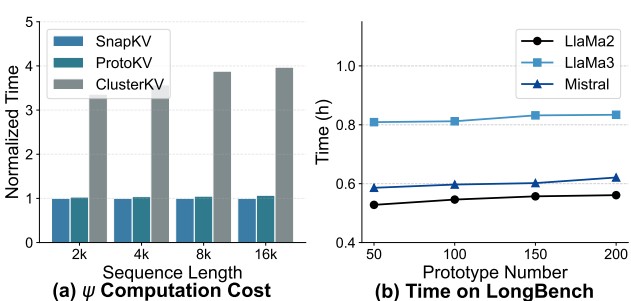

Figure 13: Computation Cost Analysis

## 6 CONCLUSION

In LLMs, we discovered an interesting phenomenon: within the key embedding space, while the vast majority of tokens exhibit a manifold distribution along input position, a small subset of special tokens which serve as semantic anchors consistently form one or several clusters. Based on this insight, we propose ProtoKV, a simplified approach to KV cache compression that leverages semantic prototypes to achieve a more rational and efficient semantic-level compression strategy.

## ACKNOWLEDGEMENTS

This work was supported in part by the National Natural Science Foundation of China (Grant Nos. 62572236,62502201,62441225), the Basic Research Program of Jiangsu Province (Grant Nos. BK20222003, BK20251198, BK20253011), the Collaborative Innovation Center of Novel Software Technology and Industrialization, and the Sino-German Institutes of Social Computing.

## ETHICS STATEMENT

The primary objective of this paper is to provide a KV cache compression framework that designed to accelerate inference. This work is based on the publicly available LongBench and Ruler dataset, which predominantly contains English text. We comply with all dataset licenses, and confirm the content contains neither private nor offensive information.

## REPRODUCIBILITY STATEMENT

Pseudocode of this paper is shown in Appendix I. In Section 5.3, we provide a hyperparameter analysis and employ the optimal set to conduct our specific experiments and obtain the final reported results. We give the link to the source code in abstract.

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

## A    USE OF LLMS

This study employs GPT-4 for text polishing and writing refinement. Moreover, we utilized Claude-3.7-Sonnet to assist us with code generation.

## B    OUTLIER PROPERTY VISUALIZATION AND ANALYSIS

We conduct a detailed investigation into the properties of **neighborhood similarity** of token key embeddings in long-text contexts. Figures 14 to 19 present results from various LLMs across different attention heads on three datasets, from which we summarize the following patterns.

**(1)** In the **initial layers** of large models, we observe a universally **high neighborhood similarity** among token key embeddings for most of the attention heads, with nearly **no outlier tokens** detected. This suggests that initial representations remain relatively homogeneous and exhibit manifold characteristics within the semantic space. We also notice that key embeddings in certain attention heads remain relatively dispersed.

**(2)** As processing proceeds to **shallower layers**, the phenomenon of **Attention Sink** begins to manifest. This is characterized by the **initial tokens** of the text aggregating global semantic information, resulting in their key embeddings exhibiting significant **outlier tendencies**. These tokens, often positioned at the beginning, absorb broad contextual attributes and become distinct from the more localized representations of other tokens.

**(3)** Subsequently in **deeper layers**, the model shifts its focus to **specific parts** of the document, leading to the emergence of **local deviation**. This means that certain **non-initial tokens** also start to exhibit outlier characteristics, particularly those relevant to specialized or salient content within the text. This progression highlights the model's evolving semantic focus across layers, from broad contextual integration to content-specific representation.

Meanwhile, we observed a very interesting phenomenon on the MK3 dataset, where the neighborhood similarity exhibits a comb-shaped distribution, meaning that an outlier token appears every few tokens. We present a sample from the MK3 dataset as follows:

> A special magic uuid is hidden within the following text. Make sure to memorize it. I will quiz you about the uuid afterwards.
> One of the special magic uuids for 42e88605-a29a-4e5f-97b6-f1aaf2064a1c is: 73e5550d-b52f-4f46-80b0-8dd0fac9b5b6.
> One of the special magic uuids for fba75385-a98a-4cb0-bef5-27fe45e4307a is: 75b568d7-35eb-428f-9775-0d0f5256e276.
> ...

It can be observed that each segment follows a key+uuid format. We found that the outliers across different heads are typically around the colons and periods, which indicates that each key-uuid pair is likely to be encoded into punctuation marks and the surrounding tokens, resulting in the periodic appearance pattern of outliers.

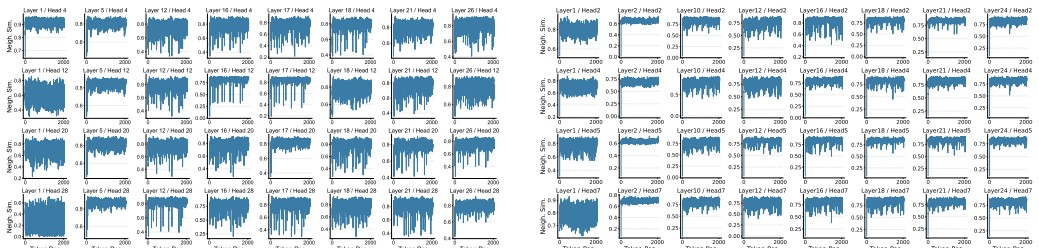

Figure 14: Token Neighborhood Similarity Across Different Attention Heads (Llama2-7B on Samsum)    Figure 15: Token Neighborhood Similarity Across Different Attention Heads (Llama3-8B on Samsum)

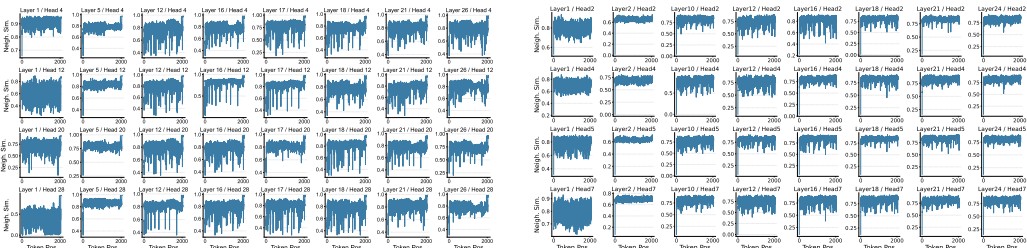

Figure 16: Token Neighborhood Similarity Across Different Attention Heads (Llama2-7B on TriviaQA)

Figure 17: Token Neighborhood Similarity Across Different Attention Heads (Llama3-8B on TriviaQA)

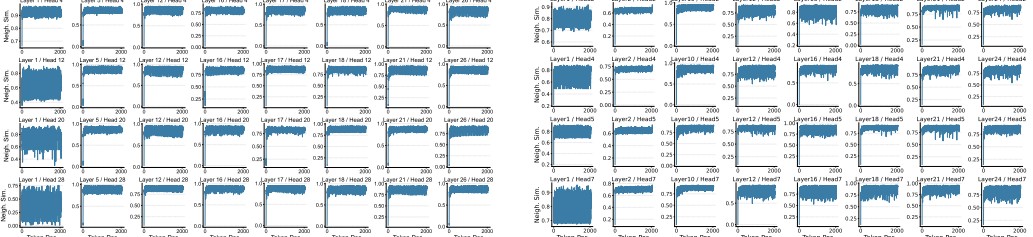

Figure 18: Token Neighborhood Similarity Across Different Attention Heads (Llama2-7B on MK3)

Figure 19: Token Neighborhood Similarity Across Different Attention Heads (Llama3-8B on MK3)

## C CLUSTERING PROPERTY ANALYSIS

In this section, we primarily investigate whether SATs exhibit clustering properties across different attention heads. Unlike the outlier property analysis experiment, where we analyzed a specific long-context sequence, here we conduct analysis on the entire dataset by calculating the clustering coefficient of key embeddings with outlier degree above the threshold $\beta$ for each input sequence, and then average them across the whole dataset.

As shown in Figures 20-23, for almost all attention heads, the clustering coefficient values show an increasing trend as $\beta$ rises, and eventually stabilize to a high value. We find that Stabilization occurs because the selected tokens are predominantly attention sinks in most cases. Additionally, we observe that the trends in the first layer are generally less pronounced, which aligns with our earlier analysis: semantic information is not yet well aggregated at the first layer.

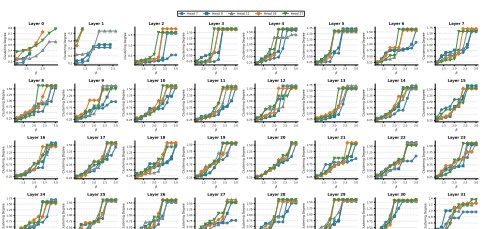

Figure 20: Clustering Property Analysis for Llama2-7B on 2WikimQA.

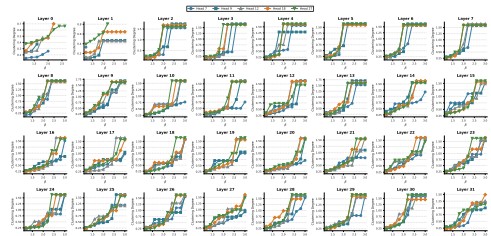

Figure 21: Clustering Property Analysis for Llama2-7B on TriviaQA.

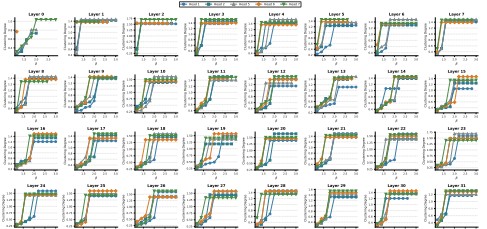

Figure 22: Clustering Property Analysis for Llama3-8B on NarrativeQA.

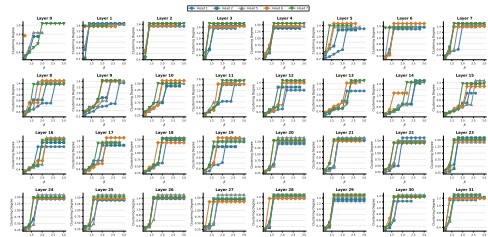

Figure 23: Clustering Property Analysis for Llama3-8B on HotpotQA.

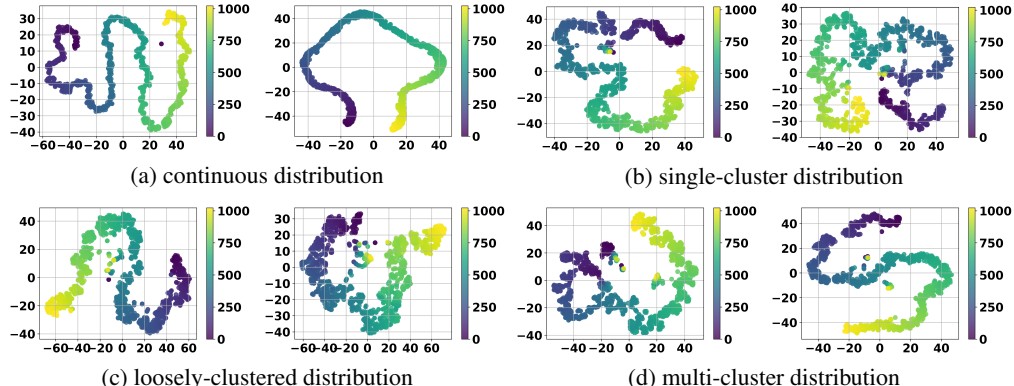

Figure 24: Diverse Clustering Pattern of Key Embeddings.

# D    KEY EMBEDDING DISTRIBUTION PATTERN ANALYSIS

In this section, we examine various distribution patterns of key embeddings, including: 1) the continuous distribution case, 2) the single-cluster distribution case, 3) the loosely-clustered distribution case, and 4) the multi-cluster distribution case. For each case, we elucidate why our proposed ProtoKV method achieves effective clustering performance.

**(1)** The continuous distribution case typically occurs in the earlier layers of the model, where semantic information has not yet been effectively aggregated. As shown in Figure 24(a), the key embeddings exhibit manifold characteristics without distinct outlier points. Under these circumstances, the semantic anchors selected via outlier degree may correspond to arbitrary tokens, and the SAT semantic prototypes may also reside in arbitrary regions. Nevertheless, the SAT semantic prototypes have negligible impact on the clustering performance, as the PDT semantic prototypes can still be effectively identified, ultimately achieving the desired chunk-partitioned clustering outcome.

**(2)** The single-cluster distribution case illustrated in Figure 24(b) is observed in the vast majority of attention heads beyond the initial layers. In this scenario, our ProtoKV method is able to identify a set of high-purity semantic anchors through outlier degree measurement. These anchor tokens are tightly clustered within a specific region of the semantic space. Consequently, even when employing a hashing-based bucketing strategy, the resulting semantic prototypes remain almost consistent. Although this approach may introduce a limited number of redundant SAT semantic prototypes, it has negligible impact on the final clustering performance.

**(3)** The loosely-clustered distribution case represents a specific variant of the single-cluster distribution scenario. As illustrated in Figure 24(c), although the SATs are indeed concentrated within a certain region of the semantic space, the clustering effectiveness remains suboptimal, exhibiting a dispersed and loosely-organized structure. Under such conditions, if only a single semantic prototype (cluster center) is selected, tokens near the cluster boundaries are likely to be misassigned to clusters governed by PDT semantic prototypes. To address this, our hashing-based bucketing strategy performs a coarse partitioning of these SATs, thereby expanding the coverage of the regions controlled by SAT semantic prototypes. This approach enhances the capability to capture higher-order semantic anchor patterns.

**(4)** The multi-cluster distribution case typically occurs when the semantics of long texts exhibit low coherence. For instance, Figure 24(d) presents a visualization of this phenomenon on the multi-document 2WikiMQA dataset. Our observations are as follows: (1) Even in multi-document datasets, only a small number of attention heads exhibit multi-cluster distributions for keys; the vast majority still adhere to either single-cluster or loosely-clustered distributions. (2) Tokens from different clusters generally originate from distinct text segments, whereas query tokens with light colors may be assigned to different clusters, reflecting that the answer may come from different text segments. In multi-cluster scenarios, it is highly reasonable for our ProtoKV to employ a hashing-based bucketing strategy to construct semantic prototypes for different SAT clusters. Moreover, since these clusters are well-separated, even a simple hashing mechanism can effectively distinguish them.

# E  SATs ARE SALIENT FOR LONG-CONTEXT INFERENCE

In this section, we provide a detailed analysis of the properties of SATs as semantic anchors. We present complete experimental results from the main text, as illustrated in Figure 26 to 29. For Llama2, it can be observed that in the first two layers, outlier tokens are not the primary focus of the attention heads. However, starting from the third layer, the model begins to attend to outlying SATs. The situation is slightly different for LlaMa3. In the LlaMa3-8B model, it can be observed that all attention heads tend to focus on outlier tokens, including those in the initial layers. Additionally, we found that even after removing the first several tokens

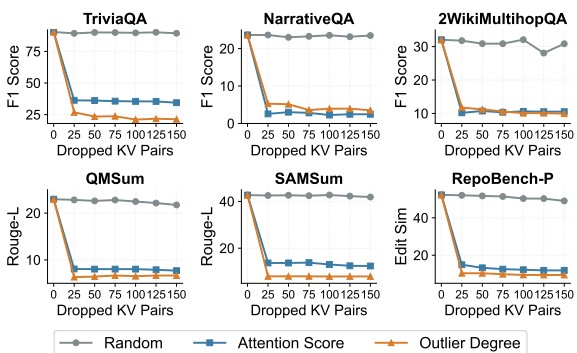

Figure 25: Dropping KV pairs with highest outlier degrees can cause sharp performance drop for llama3.

the first several tokens (i.e., the attention sink), the model still attends to the first group of tokens with the highest outlier degree, we give the results in Appendix F. This demonstrates that SAT are distributed throughout the text and commonly serve as semantic anchors.

Additionally, we also present the impact of removing the top-k KV pairs with the highest outlier degrees on Llama3's performance in processing long-text inputs (while the main text demonstrates results for Llama2). As shown in Figure 25, removing 25 KV pairs with the highest outlier degrees causes the model to almost completely malfunction. This indicates that tokens exhibiting outlier properties generally function as semantic anchors.

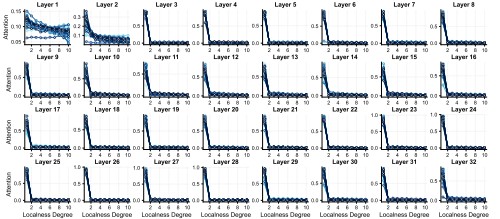

Figure 26: Group-wise semantic anchor property analysis. (Llama2-7B on TriviaQA)

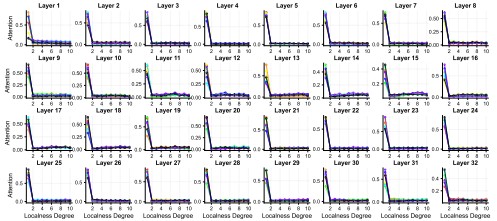

Figure 27: Group-wise semantic anchor property analysis. (LlaMa3-8B on TriviaQA)

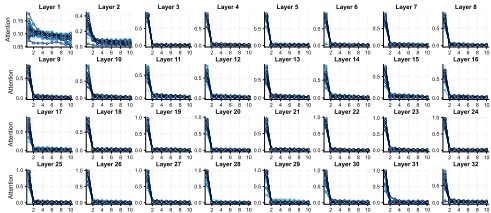

Figure 28: Group-wise semantic anchor property analysis. (Llama2-7B on Samsum)

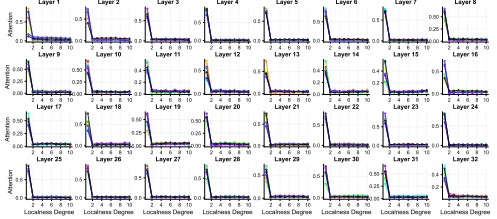

Figure 29: Group-wise semantic anchor property analysis. (LlaMa3-8B on Samsum)

# F  ALIGNMENT WITH PRIOR WORKS

We compare our findings with prior work. Specifically, (Xiao et al., 2024) observed the "attention sink" phenomenon during large model generation, where the initial tokens disproportionately absorb semantic information. In the visualization of Appendix B, we can observe that the initial few tokens often exhibit extremely low neighborhood similarity.

Furthermore, to validate whether the SATs we propose as semantic anchors are equivalent to the attention sinks mentioned in prior work, we conducted additional experiments. Specifically, we removed the first ten tokens from the long-text prompt (prior studies indicate that the first four are the most critical; here, we adopt a more aggressive approach by excluding the first ten). The

remaining tokens were then divided into ten groups based on their outlier degree, and we measured their inference-stage cumulative attention.

As illustrated in Figure 30, we observed that even after removing the attention sinks, large models still tend to focus on tokens with higher outlier degrees. This demonstrates that our SATs not only encompass attention sinks but are also capable of identifying anchors located elsewhere in the text.

(Wang et al., 2023) demonstrated that in in-context learning, label words act as anchors that cluster sample semantics. Our study reveals that **label word tokens also exhibit significant outlier characteristics**. We conduct experiments for LlaMa3-8B on two text classification datasets. These two datasets are: the binary-class movie review dataset IMDB and the multi-class news dataset AGNews.

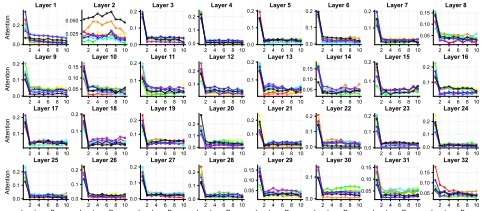

Figure 30: Dropping Attention Sink Tokens.

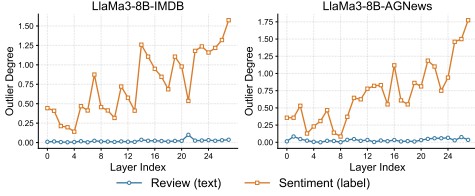

Figure 31: Label words are outliers.

The constructed prompt templates for each dataset are as follows: 1) For IMDB, the template is "Review: [text]. Sentiment: [label]," where "text" represents the movie review content and "label" indicates positive/negative sentiment. 2) For AGNews, the template is "Article: [text]. Category: [label]," where "text" represents the news article content and "label" denotes the category label. We constructed multiple sets of examples and combined them into a final prompt, which was then fed into the large language model for prefilling. Subsequently, we measured the outlier degree of different tokens in the "text" portion and the "label" portion respectively. The results, as illustrated in Figure 31, show that the outlier degree for "text" is almost zero, while the outlier degree for "labe" is generally higher and exhibits an increasing trend with deeper layers. This indicates that label words gradually become semantic anchors as layer depth increases, which aligns with findings in (Wang et al., 2023).

## G    NEEDLE IN A HAYSTACK ANALYSIS

We investigate the trends in the distribution of key embeddings of tokens in the "NEEDLE IN A HAYSTACK" experiment. First, we present the key conclusion: in retrieval heads, the neighborhood similarity of needle texts is lower than that of haystack texts, indicating that large language models (LLMs) can perceive textual incoherence and assign a higher outlier degree to incoherent texts, making them more likely to be retrieved subsequently.

Specifically, retrieval heads (Wu et al., 2024) refer to specialized attention heads in transformer-based models that are mechanistically responsible for retrieving critical information from long-context inputs. These heads dynamically activate to identify and prioritize salient contents (i.e., "needles") within extensive contexts (i.e., "haystacks"), enabling the model to maintain factual consistency and coherence in long-range dependencies. Taking Llama2 as an example, we focus on two retrieval heads identified by prior work: Head 16 in Layer 9 and Head 30 in Layer 21. As shown in Figure 32, we observe that the neighborhood similarity of needle texts in these retrieval heads is consistently lower, reinforcing their role in detecting out-of-context as salient information.

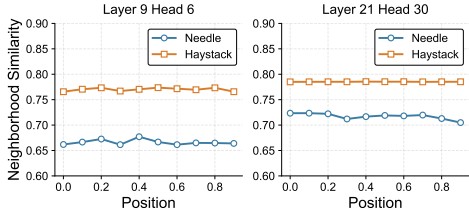

Figure 32: Needle contexts are outliers.

## H    SELECTING KV PAIRS VIA OUTLIER DEGREE

In the main text, we propose using the outlier degree as a criterion to select and retain important KV pairs, termed **SATKV**. For a long-text input of length $n$, we always retain the last $k$ KV pairs to

ensure query integrity, and select from the preceding $n - k$ KV pairs based on their outlier degree. It is noteworthy that SATKV eliminates the need for query-based attention computation; instead, it achieves effective KV compression using keys alone. This implies that SATKV's compression results may exhibit universality across diverse queries. Moreover, since $\kappa$ is typically smaller than both the budget $\mathcal{B}$ and the observation window size, SATKV operates more efficiently.

Table 5: Time Complexity Comparison

| LLM | H2O | SnapKV | SATKV |
|------|------|--------|-------|
| $\mathcal{O}(1)$ | $\mathcal{O}(n\mathcal{B}d)$ | $\mathcal{O}(n\mathcal{L}_{obs}d)$ | $\mathcal{O}(n\kappa d)$ |

## I PSEUDOCODE

We present the pseudo-code for our ProtoKV as follows, it is worth noting that all loop statements in this code can be executed in parallel. We analyze the time complexity for it, assuming a key dimension of $d$. Computing the $\kappa$-Neighborhood Similarity to obtain the SAT requires $\mathcal{O}(n\kappa d)$ time. Constructing the hybrid semantic prototypes has a complexity of $\mathcal{O}(nd)$, while assigning prototypes to each token incurs $\mathcal{O}(nkd)$ time. Finally, the observation window selection step takes $\mathcal{O}(nL_{obs}d)$ time. Since $k$ is typically larger than both $L_{obs}$ and $\kappa$ in practical experiments, the overall time complexity can be approximated as $\mathcal{O}(nkd)$.

---

**Algorithm 1** Algorithm of ProtoKV

---

**Require:** Key vectors $\{\mathbf{k}_t\}_{t=1}^n$, chunk size $v$, hash bits $r$, buckets $u$, SAT selected number $p$, observation window size $L_{obs}$, attention head number $H$, Key-Value budget size $\mathcal{B}$
**Ensure:** Compressed KV cache $\{\tilde{K}^{(h)}, \tilde{V}^{(h)}\}_{h=1}^H$
 1: **for** each head $h \in [H]$ **do**
 2:     Calculate $\Theta^{(h)}(i)$ according to Eq. 5
 3:     $\mathcal{O} \leftarrow$ Top-$p$ of $\Theta^{(h)}(i)$
 4:     **for** $m = 1$ **to** $v$ **do**
 5:         $\mathcal{C}_m \leftarrow \{\mathbf{k}_{(m-1)\lfloor n/v \rfloor + 1}, ..., \mathbf{k}_{m\lfloor nv \rfloor}\}$
 6:         $c_m^{(\text{PDTs})} \leftarrow \frac{\sum_{\mathbf{k}_t \in \mathcal{C}_m \backslash \mathcal{O}} \mathbf{k}_t}{|\mathcal{C}_m \backslash \mathcal{O}|}$
 7:     **end for**
 8:     **for** $\mathbf{k}_j \in \mathcal{O}$ **do**
 9:         $\phi(\mathbf{k}_j) \leftarrow \sqrt{2/r} \cos(W\mathbf{k}_j + b)$
10:         $\mathcal{H}(\mathbf{k}_j) \leftarrow \text{Binarize}(\phi(\mathbf{k}_j))$
11:     **end for**
12:     **for** $s = 1$ **to** $u$ **do**
13:         $\mathcal{B}_s \leftarrow \{\mathbf{k}_j | \mathcal{H}(\mathbf{k}_j) = s\}$
14:         $c_s^{(\text{SATs})} \leftarrow \frac{\sum_{\mathbf{k}_j \in \mathcal{B}_s} \mathbf{k}_j}{|\mathcal{B}_s|}$
15:     **end for**
16:     $\mathcal{M} \leftarrow \{c_m^{(\text{PDTs})}\}_{m=1}^v \cup \{c_s^{(\text{SATs})}\}_{s=1}^u$
17:     **for** each token $\mathbf{k}_t$ **do**
18:         $\mathcal{C}(\mathbf{k}_t) \leftarrow \arg\max_{c \in \mathcal{M}} \frac{\mathbf{k}_t^\top c}{\|\mathbf{k}_t\|_2 \|c\|_2}$
19:     **end for**
20:     **for** $i = 1$ **to** $n$ **do**
21:         $S^{(h)}(i) \leftarrow \sum_{m=L_{\text{prefix}}+1}^{L_{\text{prompt}}} \mathbf{q}_m^\top \mathbf{k}_i$
22:         $\hat{S}^{(h)}(i) \leftarrow \frac{1}{|\{t|C(\mathbf{k}_t)=C(k_i)\}|} \sum_{C(\mathbf{k}_t)=C(\mathbf{k}_i)} S^{(h)}(t)$
23:     **end for**
24:     $I^{(h)} \leftarrow$ Top-$(\mathcal{B} - L_{obs})$ indices of $\hat{S}^{(h)}$
25:     $\tilde{K}^{(h)} \leftarrow \{\mathbf{k}_i^{(h)}\}_{i \in I^{(h)}} \cup \{\mathbf{k}_j^{(h)}\}_{j=L_{\text{prefix}}+1}^{L_{\text{prompt}}}$
26:     $\tilde{V}^{(h)} \leftarrow \{\mathbf{v}_i^{(h)}\}_{i \in I^{(h)}} \cup \{\mathbf{v}_j^{(h)}\}_{j=L_{\text{prefix}}+1}^{L_{\text{prompt}}}$
27: **end for**

---

## J LONGBENCH DATASET DETAILS

**Dataset** LongBench is a large-scale benchmark dataset designed for evaluating language models' capabilities in understanding and generating long texts. It covers various types of tasks including, but not limited to, Single-Document Question Answering (QA), Multi-Document QA, Summarization, Few-shot Learning, and Synthetic tasks. The aim is to comprehensively assess models across different application scenarios.

Here are some specific tasks included in the LongBench dataset along with their characteristics:

**NarrativeQA**: Focuses on understanding narrative texts, requiring models to read and answer questions about stories or narratives.

**Qasper**: Involves asking and answering questions based on academic articles, testing the model's ability to understand scholarly literature.

**MultiFieldQA-en**: Covers QA tasks across multiple fields, enhancing the model's capability to understand texts from diverse domains.

**HotpotQA, 2WikiMultihopQA, MuSiQue**: These tasks emphasize reasoning and information integration across multiple documents, challenging the model's ability to find answers in a multi-document environment.

**GovReport, QMSum, MultiNews**: Concentrate on extracting key information and generating summaries from lengthy texts, assessing the model's summarization capability.

**TREC, TriviaQA, SAMSum**: Evaluate the model's learning ability and domain-specific knowledge acquisition through few-shot examples.

**PassageCount, PassageRetrieval-en**: Synthetic tasks designed to test the model's performance under specific conditions, such as document counting or retrieval accuracy.

**LCC, RepoBench-P**: Involve code understanding and evaluation of editing similarity, catering to the unique requirements of programming languages.

Each task comes with its own set of evaluation metrics (e.g., F1 Score, Rouge-L, Accuracy) to quantify model performance. Moreover, LongBench includes texts from different languages and domains, ensuring broad applicability and linguistic diversity of the models. This dataset plays a crucial role in advancing the field of natural language processing, especially in improving models' abilities to handle long texts. Detailed information is demonstrated in Table 6.

| Task | Task Type | Source | Eval metric | Avg len | Language | License |
|------|-----------|--------|-------------|---------|----------|---------|
| NarrativeQA | Single-Doc. QA | Literature, Film | F1 | 18,409 | EN | MIT License |
| Qasper | Single-Doc. QA | Science | F1 | 3,619 | EN | MIT License |
| MultiFieldQA-en | Single-Doc. QA | Multi-field | F1 | 4,559 | EN | MIT License |
| HotpotQA | Multi-Doc. QA | Wikipedia | F1 | 9,151 | EN | MIT License |
| 2WikiMultihopQA | Multi-Doc. QA | Wikipedia | F1 | 4,887 | EN | MIT License |
| MuSiQue | Multi-Doc. QA | Wikipedia | F1 | 11,214 | EN | MIT License |
| GovReport | Summarization | Government report | Rouge-L | 8,734 | EN | MIT License |
| QMSum | Summarization | Meeting | Rouge-L | 10,614 | EN | MIT License |
| MultiNews | Summarization | News | Rouge-L | 2,113 | EN | MIT License |
| TREC | Few shot | Web question | Accuracy | 5,177 | EN | MIT License |
| TriviaQA | Few shot | Wikipedia, Web | F1 | 8,209 | EN | MIT License |
| SAMSum | Few shot | Dialogue | Rouge-L | 6,258 | EN | MIT License |
| PassageCount | Synthetic | Wikipedia | Accuracy | 11,141 | EN | MIT License |
| PassageRetrieval-en | Synthetic | Wikipedia | Accuracy | 9,289 | EN | MIT License |
| LCC | Code | Github | Edit Sim | 1,235 | Python/C#/Java | MIT License |
| RepoBench-P | Code | Github | Edit Sim | 4,206 | Python/Java | MIT License |

Table 6: An overview of the dataset statistics in LongBench.

## K LLM MODEL DETAILS

**Meta-Llama-3-8B-Instruct** is an 8B-parameter instruction-tuned variant of LLaMA-3, optimized for dialogue tasks. Using transformer architecture with SFT and RLHF, it features a 128K vocabulary and GQA for efficiency. The model supports 8K-context (extendable to 128K) and demonstrates strong performance in text generation and reasoning tasks.

**Mistral-7B-Instruct-v0.2** is a 7.3B-parameter instruction-tuned model by Mistral AI, featuring 32K context length via optimized RoPE embeddings. With grouped-query attention for efficiency, it excels in conversational and coding tasks while supporting GGUF quantization. Benchmarks show it outperforms comparable 7B models, particularly in code generation.

**Llama2-7B-chat** is Meta's 7 billion parameter chat-optimized language model, fine-tuned for dialogue applications using RLHF. The model features a 4K token context window and demonstrates improved safety and helpfulness compared to its base version. It achieves strong performance in conversational tasks while maintaining efficient inference through optimized transformer architecture.

| Configuration | LlaMA-3-8B-Instruct | Mistral-7B-Instruct-v0.2 | Llama2-7B-chat |
|---|---|---|---|
| Hidden Size | 4,096 | 4,096 | 4,096 |
| Layers | 32 | 32 | 32 |
| Q Heads | 32 | 32 | 32 |
| KV Heads | 8 | 8 | 32 |
| Attention Heads | 32 | 32 | 32 |
| Max Position Embeddings | 8,192 | 32,768 | 4,096 |
| Intermediate Size | 14,336 | 14,336 | 11,008 |
| Vocabulary Size | 128,256 | 32,000 | 32,000 |

Table 7: Configuration of Models.

## L  DETAILED RESULTS FOR LONGBENCH

| Model | Size | Method | Single-Document QA | | | Multi-Document QA | | | Summarization | | | Few-shot Learning | | | Synthetic | | Code | | Avg. |
|---|---|---|---|---|---|---|---|---|---|---|---|---|---|---|---|---|---|---|---|
| | | | NrtvQA | Qasper | MF-en | HotpotQA | 2WikiMQA | Musique | GovReport | QMSum | MultiNews | TREC | TriviaQA | SAMSum | PCount | PRe | Lcc | RB-P | |
| | | | 18409 | 3619 | 4559 | 9151 | 4887 | 11214 | 8734 | 10614 | 2113 | 5177 | 8209 | 6258 | 11141 | 9289 | 1235 | 4206 | – |
| LLaMA-3-8B-Instruct | – | FullKV | 25.16 | 32.29 | 40.43 | 45.35 | 37.04 | 23.84 | 28.62 | 23.34 | 26.33 | 75.00 | 90.23 | 42.65 | 5.10 | 70.0 | 59.41 | 55.60 | 42.34 |
| | 128 | SLM | 17.47 | 8.55 | 21.31 | 32.86 | 26.28 | 15.54 | 17.91 | 20.42 | 20.16 | 45.00 | 73.36 | 30.78 | 5.75 | 68.50 | 48.38 | 49.31 | 31.29 |
| | | H2O | 21.58 | 12.54 | 28.57 | 39.86 | 28.62 | 18.88 | 20.23 | 22.16 | 20.14 | 35.50 | 86.62 | 39.19 | 5.83 | 69.50 | 54.46 | 50.81 | 34.66 |
| | | SnapKV | 22.35 | 16.00 | 31.52 | 36.82 | 28.39 | 19.49 | 19.06 | 21.36 | 20.07 | 50.00 | 87.74 | 38.94 | 5.75 | 68.00 | 57.42 | 51.84 | 35.92 |
| | | ChunkKV | 23.08 | 16.32 | 30.18 | 37.25 | 28.60 | 19.12 | 19.49 | 20.45 | 20.45 | 53.00 | 88.25 | 39.57 | 5.81 | 68.00 | 58.13 | 53.73 | 36.36 |
| | | Pyramid | 21.80 | 16.65 | 30.73 | 38.48 | 28.80 | 19.26 | 19.92 | 22.06 | 20.12 | 66.50 | 88.95 | 38.20 | 5.92 | 68.00 | 57.88 | 51.54 | 37.16 |
| | | ProtoKV | 22.26 | 17.05 | 31.84 | 39.68 | 29.28 | 19.35 | 19.83 | 22.31 | 20.82 | 62.00 | 89.35 | 38.74 | 5.37 | 69.00 | 58.84 | 54.61 | 37.52 |
| | 256 | SLM | 17.98 | 11.09 | 23.85 | 37.83 | 29.97 | 16.02 | 20.30 | 20.94 | 24.56 | 52.00 | 79.68 | 34.82 | 5.83 | 69.50 | 54.84 | 50.46 | 34.30 |
| | | H2O | 23.67 | 16.85 | 32.70 | 41.57 | 31.08 | 18.91 | 22.28 | 22.81 | 23.69 | 41.00 | 90.36 | 40.19 | 5.54 | 69.50 | 57.52 | 52.16 | 36.85 |
| | | SnapKV | 23.32 | 20.31 | 37.35 | 42.70 | 31.08 | 20.47 | 22.63 | 23.04 | 23.93 | 71.00 | 90.39 | 39.78 | 5.50 | 69.50 | 60.27 | 55.62 | 39.81 |
| | | ChunkKV | 23.49 | 20.12 | 37.01 | 42.95 | 31.32 | 20.68 | 22.94 | 23.02 | 23.68 | 70.00 | 90.60 | 39.93 | 5.65 | 69.50 | 60.75 | 55.56 | 39.83 |
| | | Pyramid | 23.46 | 18.76 | 35.06 | 42.33 | 31.56 | 20.73 | 23.37 | 23.11 | 24.37 | 72.00 | 90.43 | 39.54 | 5.50 | 69.50 | 59.25 | 54.87 | 39.61 |
| | | ProtoKV | 23.58 | 19.92 | 36.38 | 43.72 | 32.29 | 20.89 | 23.25 | 22.98 | 23.42 | 70.00 | 90.81 | 40.07 | 5.80 | 69.50 | 61.22 | 55.49 | 39.95 |
| Mistral-7B-Instruct | – | FullKV | 25.07 | 32.92 | 49.34 | 39.77 | 27.32 | 16.83 | 32.87 | 24.24 | 27.10 | 70.00 | 86.57 | 43.30 | 2.75 | 59.25 | 56.86 | 50.48 | 40.29 |
| | 128 | SLM | 17.76 | 13.46 | 35.11 | 27.25 | 22.29 | 9.80 | 18.26 | 19.02 | 19.16 | 43.50 | 74.12 | 36.50 | 2.67 | 27.17 | 43.65 | 43.79 | 28.34 |
| | | H2O | 19.99 | 20.34 | 38.60 | 28.50 | 21.63 | 12.88 | 20.65 | 22.61 | 22.08 | 53.00 | 81.29 | 39.75 | 2.20 | 75.38 | 49.54 | 44.27 | 33.83 |
| | | SnapKV | 22.14 | 21.14 | 42.98 | 32.96 | 22.12 | 14.12 | 19.19 | 21.89 | 21.01 | 64.00 | 83.77 | 39.92 | 2.51 | 66.50 | 51.81 | 46.51 | 35.84 |
| | | ChunkKV | 22.13 | 22.04 | 43.62 | 33.52 | 22.28 | 14.29 | 20.38 | 22.95 | 21.73 | 65.25 | 82.33 | 40.18 | 2.63 | 69.23 | 51.68 | 45.23 | 36.22 |
| | | Pyramid | 22.32 | 22.52 | 43.65 | 33.07 | 22.45 | 15.72 | 20.56 | 22.52 | 21.36 | 64.00 | 83.84 | 40.43 | 2.74 | 67.95 | 51.64 | 46.47 | 36.29 |
| | | ProtoKV | 23.11 | 23.70 | 44.89 | 36.12 | 22.88 | 15.75 | 21.63 | 23.75 | 22.49 | 67.50 | 84.89 | 41.96 | 3.10 | 72.30 | 53.54 | 47.95 | 37.84 |
| | 256 | SLM | 19.26 | 17.78 | 36.82 | 27.74 | 22.78 | 10.53 | 24.47 | 19.84 | 25.48 | 51.00 | 76.39 | 40.24 | 2.50 | 31.92 | 46.15 | 45.56 | 31.14 |
| | | H2O | 22.35 | 23.22 | 41.76 | 30.76 | 22.88 | 14.03 | 23.53 | 22.96 | 24.53 | 53.50 | 83.82 | 41.08 | 1.66 | 78.49 | 50.77 | 46.70 | 36.39 |
| | | SnapKV | 23.08 | 25.95 | 48.04 | 34.79 | 24.75 | 14.41 | 24.14 | 23.69 | 24.47 | 67.50 | 85.64 | 41.51 | 1.95 | 68.11 | 53.74 | 49.31 | 38.19 |
| | | ChunkKV | 23.60 | 26.22 | 48.53 | 35.16 | 25.45 | 14.53 | 24.41 | 23.52 | 24.13 | 66.50 | 84.81 | 41.42 | 2.60 | 70.25 | 53.82 | 50.64 | 38.54 |
| | | Pyramid | 23.49 | 26.39 | 48.22 | 35.23 | 25.51 | 13.65 | 24.79 | 23.52 | 24.49 | 68.50 | 85.43 | 41.58 | 2.33 | 69.07 | 53.45 | 48.23 | 38.37 |
| | | ProtoKV | 23.76 | 26.02 | 48.82 | 34.96 | 26.32 | 14.66 | 24.69 | 23.62 | 24.91 | 70.50 | 86.02 | 42.76 | 2.90 | 71.40 | 55.62 | 51.14 | 39.25 |
| Llama2-7B-chat | – | Full | 14.82 | 9.5 | 22.76 | 7.35 | 10.71 | 9.23 | 25.63 | 23.79 | 26.51 | 65.00 | 89.16 | 34.28 | 2.50 | 9.50 | 68.24 | 61.83 | 29.72 |
| | 128 | SLM | 10.12 | 4.94 | 15.8 | 5.93 | 9.07 | 3.05 | 18.07 | 19.30 | 18.30 | 42.50 | 76.97 | 24.18 | 2.00 | 3.11 | 61.47 | 44.26 | 22.93 |
| | | H2O | 13.22 | 4.55 | 16.28 | 6.58 | 9.01 | 3.82 | 20.92 | 21.86 | 18.44 | 40.00 | 79.40 | 27.85 | 1.20 | 7.38 | 55.75 | 53.36 | 24.09 |
| | | SnapKV | 13.13 | 5.84 | 21.62 | 7.12 | 9.19 | 3.90 | 18.91 | 21.41 | 18.21 | 45.00 | 84.12 | 27.85 | 1.60 | 7.02 | 61.48 | 54.87 | 25.62 |
| | | ChunkKV | 13.42 | 5.81 | 21.80 | 7.47 | 9.02 | 3.86 | 19.60 | 21.42 | 18.69 | 51.00 | 84.32 | 28.72 | 1.60 | 7.95 | 62.80 | 55.47 | 25.81 |
| | | Pyramid | 13.78 | 5.75 | 22.37 | 7.62 | 9.68 | 3.96 | 19.24 | 20.47 | 18.18 | 59.00 | 84.38 | 29.42 | 1.50 | 8.22 | 62.24 | 54.51 | 26.45 |
| | | ProtoKV | 13.97 | 5.94 | 22.08 | 7.76 | 9.29 | 4.09 | 20.85 | 21.60 | 19.02 | 59.50 | 84.69 | 29.99 | 1.50 | 8.18 | 63.22 | 56.97 | 26.94 |
| | 256 | SLM | 12.74 | 4.94 | 15.8 | 5.93 | 9.12 | 3.48 | 25.70 | 19.31 | 24.87 | 54.00 | 81.67 | 31.47 | 2.00 | 4.38 | 61.87 | 52.20 | 25.60 |
| | | H2O | 14.55 | 5.95 | 18.67 | 6.42 | 8.67 | 4.17 | 23.69 | 22.07 | 22.72 | 56.00 | 82.66 | 30.48 | 2.50 | 8.89 | 58.83 | 56.83 | 26.45 |
| | | SnapKV | 17.12 | 6.75 | 21.52 | 7.38 | 10.03 | 4.12 | 24.56 | 22.39 | 23.07 | 63.00 | 84.96 | 31.54 | 1.52 | 7.25 | 64.94 | 56.88 | 28.01 |
| | | ChunkKV | 17.56 | 7.03 | 21.29 | 7.26 | 10.22 | 4.47 | 24.15 | 22.75 | 23.15 | 65.00 | 85.52 | 32.17 | 1.68 | 8.54 | 65.30 | 58.12 | 28.39 |
| | | Pyramid | 17.84 | 7.28 | 20.37 | 7.14 | 10.47 | 4.29 | 23.59 | 22.30 | 22.41 | 64.00 | 85.17 | 32.72 | 2.67 | 8.23 | 65.75 | 57.50 | 28.30 |
| | | ProtoKV | 17.29 | 7.34 | 20.94 | 7.58 | 11.43 | 4.80 | 24.73 | 22.57 | 22.89 | 68.00 | 86.78 | 34.51 | 1.68 | 7.67 | 66.88 | 60.34 | 29.11 |

Figure 33: Performance comparison on the **LongBench** dataset for full KV cache, extant KV baselines (including StreamingLLM, H2O, SnapKV, ChunkKV, PyramidKV) and our ProtoKV.

## M RELATED WORKS

**KV Cache Compression** focuses on retaining critical key-value pairs while permanently discarding unimportant ones to optimize memory and inference. Two dominant strategies emerge: (1) static methods with prefill-phase token selection (Ge et al., 2024; Li et al., 2024b; Zeng et al., 2024), and (2) dynamic approaches updating cached entries via attention-based metrics or structural patterns during decoding (Xiao et al., 2024; Han et al., 2024; Zhang et al., 2023; Zhao et al., 2024). Recent advancements address persistent eviction challenges through multi-tier caching and asynchronous retrieval (Lee et al., 2024; Tang et al., 2024; Zhang et al., 2024a; Hooper et al., 2024a; Liu et al., 2024a). However, most of these solutions fail to efficiently preserve semantic coherence, leading to suboptimal selection decisions.

**Semantic-Level KV Cache Selection** Optimizing KV cache at a semantic level is crucial to maintaining output coherence. (Li et al., 2024b; Liu et al., 2025a) group tokens into semantic chunks, retaining the most informative segments and discarding redundant ones to enhance long-context inference efficiency. Recently, clustering-based KV compression approaches (Liu et al., 2024b; Hooper et al., 2024a) clusters tokens semantically and recalls them at the granularity of semantic clusters.

**Discussion: Why Clustering?** In our view, clustering-based KV compression can be explained via graph neural networks (GNNs) Kipf & Welling (2017); Hong et al. (2025); Lin et al. (2025); Yu et al. (2025); Lin et al. (2023), as the two are closely related. In GNNs, a node's importance is influenced by its neighbors, similarly, a KV pair's importance comes from aggregating its neighboring KVs. SnapKV defines neighbors as fixed-window tokens and uses max pooling, while clustering-based methods group tokens by key similarity and apply mean pooling, which is more reasonable. However, like GNNs, clustering-based methods also face scalability challenges Lin et al. (2022); Hong et al. (2024); Lin et al. (2024).

**KV Cache Budget Allocation** LLMs' hierarchical layers exhibit distinct information extraction patterns, motivating adaptive memory allocation across layers/heads. Layer-wise strategies (Cai. et al., 2024; Yang et al., 2024; Huang et al., 2024; Zhang et al., 2024b) prioritize resource distribution by analyzing attention concentration gradients, where lower layers retain uniform contextual signals while higher layers preserve semantic focal points. Head-wise approaches (Feng et al., 2024; Zhang et al., 2024c; Fu et al., 2024b) further enable finer-grained optimization through intra-layer importance differentiation.

## N FURTHER ANALYSIS FOR LOCAL DEVIATION PROPERTY

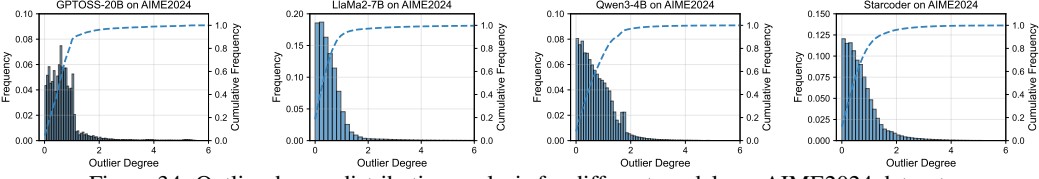

Figure 34: Outlier degree distribution analysis for different models on AIME2024 dataset.

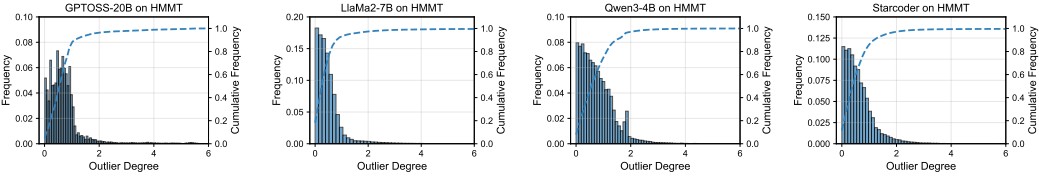

Figure 35: Outlier degree distribution analysis for different models on HMMT dataset.

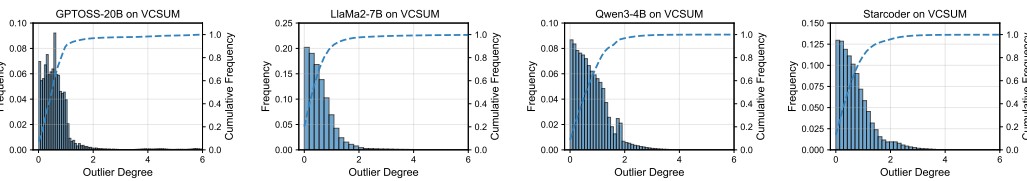

Figure 36: Outlier degree distribution analysis for different models on VCSUM dataset.

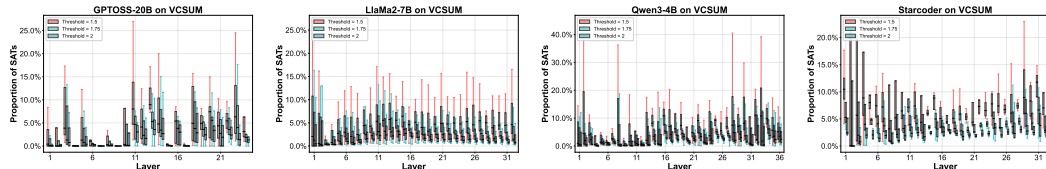

Figure 37: Layer-wise distribution analysis for SATs across different models using VCSUM dataset.

In this section, we further validate the widespread presence of the **Local Deviation Property** across diverse large language model architectures and various types of datasets. To this end, we examine representative models from three major attention architecture families: the original **multi-head attention** (MHA), as represented by Llama2-7B; **grouped-query attention** (GQA), represented by the recent Qwen3-4B; and **multi-query attention** (MQA), which is less commonly used today and for which we employ the earlier model Starcoder. Additionally, we test GPTOSS-20B, an open-source mixture-of-experts model with **flash-attention-with-sink implements**.

In terms of datasets, alongside LongBench used in the main text, we conduct key distribution analyses on the AIME-2024 and HMMT datasets, both of which are mathematical problem datasets. For instance, AIME comprises challenging problems from the American Invitational Mathematics Examination, characterized by rigorous, logically sound solutions with minimal noise. Each problem is formatted as *"Problem: [...], Solution: [...]"*, and then fed into to the large language model for prefilling. After that, we extract the key vectors head-wise, and analyze the outlier degree distribution of them.

Our experimental results demonstrate that:

- The **Local Deviation Property** persists across all tested attention architectures, including the mixture-of-experts model. This is reflected in the presence of tokens with high outlier degrees when processing various types of long-context data, as visualized in Figures 34-36.

- The phenomenon is observed consistently across both long-context benchmarks (LongBench and RULER) and mathematical reasoning datasets. Moreover, the density distribution of outlier degrees appears to be model-dependent rather than dataset-specific.

- We also provide layer-wise distributions of SATs in Figure 37. Specifically, we quantify the number of outlier keys per KV head within each layer and visualize their number distributions using box plots, with SAT detection thresholds set at 1.5, 1.75, and 2. The results indicate that th'e majority of KV heads contain outlier key vectors (i.e., SATs), which aligns fully with the conclusions presented in the main text.

## O DYNAMIC DECODING DESIGN

Different from focusing on the prefilling stage, **Dynamic Decoding Design** focuses on the management of KV cache during the reasoning stage of LLM. Specifically, we allocate memory for two components: a cache of budget size $B_{\text{budget}}$ to store retained KV tokens, and a buffer of size $B_{\text{buffer}}$ for newly generated text tokens. The total memory requirement is thus $B_{\text{total}} = B_{\text{budget}} + B_{\text{buffer}}$. After the model generates each fixed-length text segment in the buffer, R-KV performs KV cache compression. At the end of each text segment, the last $\alpha$ tokens are always retained in the cache as **observation tokens**. Next, we concatenate the existing $B_{\text{budget}}$ tokens in the cache with the first $B_{\text{buffer}} - \alpha$ tokens in the buffer, resulting in $n = B_{\text{budget}} + B_{\text{buffer}} - \alpha$ candidate KV tokens. Each candidate is assigned a selection score, and we select the top $k = B_{\text{budget}} - \alpha$ tokens to fit in the rest of the cache budget, in addition to the $\alpha$ observation tokens. This process compresses the KV cache while preserving critical context, enabling efficient memory utilization during autoregressive decoding.

Table 8: Performance (%) for dynamic decoding setting with **Llama3-8B**.

| Benchmark | Method | KV Budget | |
|---|---|---|---|
| | | 1024 | 2048 |
| MATH | FullKV | 82.38 | 82.38 |
| | H2O | 71.64 | 78.15 |
| | SnapKV | 74.43 | 80.50 |
| | ChunkKV | 76.69 | 80.42 |
| | ProtoKV | 79.67 | 81.39 |
| AIME | FullKV | 49.79 | 49.79 |
| | H2O | 19.51 | 29.87 |
| | SnapKV | 17.73 | 32.76 |
| | ChunkKV | 18.32 | 34.98 |
| | ProtoKV | 21.05 | 38.26 |

We evaluate the performance of different KV Cache compression algorithm on the MATH and AMIE dataset, both of which are mathematical problem datasets. The results are presented in Table 8. As shown, our method achieves consistent improvements over existing compression techniques. For instance, under a KV budget of 1024, ProtoKV outperforms H2O by 8.03%, SnapKV by 5.24%, and ChunkKV by 2.98%. When the KV budget increases to 2048, ProtoKV still maintains a clear advantage. On the more challenging AIME benchmark, ProtoKV demonstrates even more significant advantages. With a 1024 KV budget, it surpasses H2O, SnapKV, and ChunkKV by 7.89%, 18.73%, and 14.90% respectively. When the KV budget expands to 2048, the performance gap further widens, with ProtoKV achieving improvements of 8.39% over H2O, 5.50% over SnapKV, and 3.28% over ChunkKV. These observations indicate that: ProtoKV remains effective not only during the prefilling stage but also under dynamic decoding scenarios.

## P    ADDITIONAL EXPERIMENTAL RESULTS ON RULER

In the main text, we evaluated various KV compression methods using the RULER benchmark with a 4K context length. To further assess the scalability of these methods under more demanding conditions, we extend the evaluation to longer contexts of 16K and 32K. We report the performance of Llama3-8B with a budget size of 512 under different KV compression approaches. As detailed in Table 9, our proposed ProtoKV achieves superior performance, attaining 68.3% and 57.2% on the 16K and 32K contexts, respectively. This represents a substantial margin over competing methods. Notably, ProtoKV outperforms H2O remarkably under the 16K and 32K settings. When compared to other strong baselines, ProtoKV maintains

|         | Context | |
|---------|------|------|
| Method  | 16K  | 32K  |
| FullKV  | 85.7 | 79.9 |
| H2O     | 27.4 | 24.6 |
| SnapKV  | 64.0 | 53.4 |
| ChunkKV | 65.9 | 54.7 |
| PyramidKV | 64.2 | 53.9 |
| ProtoKV | 68.3 | 57.2 |

Table 9: Performance (%) on Ruler for Llama3-8B with budget size 512.

consistent and non-trivial improvements: 6.7-7.1% over SnapKV, 3.6-4.6% over ChunkKV, and 6.1-6.4% over PyramidKV. While the performance gain of ProtoKV over other advanced methods was marginal in the 4K context setting discussed in the main text, these new results demonstrate that its advantage becomes pronounced and increasingly significant as the context length extends, highlighting its superior scalability and effectiveness in long-context scenarios.

## Q    CONFIDENCE INTERVALS ANALYSIS WITH TEMPERATURE SAMPLING

To more rigorously validate that the performance improvements of our method are robust and not merely attributable to the inherent stochasticity of large language models, we for the first time conduct a significance analysis of different KV cache algorithms. Specifically, we configure the model to perform sampling during the decoding phase, with a temperature of 0.6 and a top-k value of 20 (i.e., sampling is constrained to the top 20 tokens in the vocabulary at each generation step). We perform five decoding runs on each sub-task dataset in LongBench and report the mean and standard deviation for LlaMa3-8B.

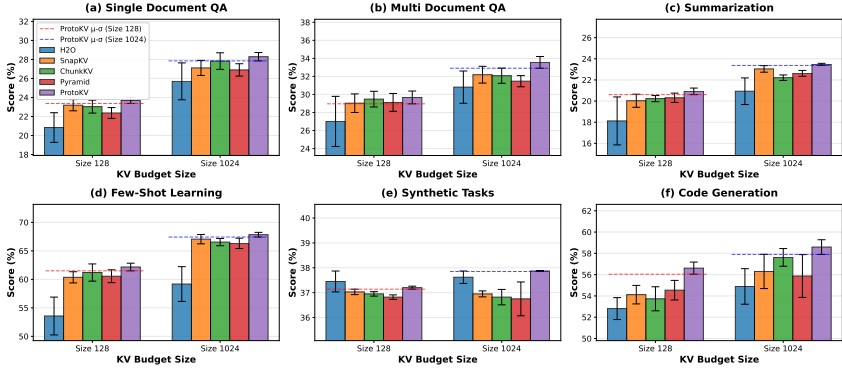

Figure 38: Results for Temperature Sampling.

We present the experimental results in Figure 38. The analysis reveals three key findings: First, our approach consistently achieves state-of-the-art performance across various subtask benchmarks. Second, in specific domains such as Summarization, the lower confidence bound of our method exceeds the upper bounds of competing baselines, demonstrating that the observed improvements are statistically significant and not merely artifacts of LLM stochasticity. Third, our method exhibits the smallest variance in performance scores, indicating enhanced robustness and more reliable model outputs compared to existing techniques.

## R    COULD PROTOKV ACHIEVES FULLKV'S LEVEL OF PRECISION?

In our preliminary experiments, we observed a discernible performance gap between our proposed ProtoKV and the FullKV baseline. To investigate whether ProtoKV can attain the precision level of FullKV under an increased budget allocation, we conducted additional experiments on LongBench using LlaMa3-8B. Specifically, we adhered to the experimental protocol of ChunkKV and evaluated performance using budget percentages relative to the total context size. As illustrated in Table 10, given LongBench's average input length of approximately 10K tokens, a 20% budget—corresponding to roughly 2048 tokens, which suffices to nearly match FullKV's performance. Moreover, with a 30% budget, ProtoKV slightly surpasses FullKV, which we attribute to the selective eviction of certain KV cache entries that helps the model focus more effectively on task-relevant text segments while reducing attention to less relevant information.

Table 10: Average score on Longbench for Llama-3B.

|  | 10% Budget | 20% Budget | 30% Budget |
| --- | --- | --- | --- |
| *FullKV* | 42.34 | 42.34 | 42.34 |
| *ProtoKV* | 41.66 | 42.27 | 42.49 |
| Improvement | -1.61% | -0.16% | +0.37% |

## S    EXPERIMENTS FOR MORE RECENT LLMS

In addition to the three large language models discussed in the main text, we have extended our evaluation to include two more recent and advanced models: Phi-3.5-mini-instruct and Mistral-7B-Instruct-v0.3.

Phi-3.5-mini is a lightweight, state-of-the-art open model constructed from the dataset used for Phi-3, which incorporates synthetic data and filtered publicly available web data, with a strong emphasis on high-quality, reasoning-intensive information. As a member of the Phi-3 model family, it supports an extensive context length of 128K tokens. The model has undergone a rigorous enhancement process that combines supervised fine-tuning, proximal policy optimization, and direct preference optimization to ensure precise instruction adherence and robust safety measures.

Mistral-7B-Instruct-v0.3 is a 7-billion parameter instruction-tuned language model developed by Mistral AI. It utilizes advanced architectural features such as Grouped-Query Attention and Sliding-Window Attention, which enhance computational efficiency and performance across various tasks including question answering, text generation, summarization, and logical reasoning. The model is fully open-source and designed for efficient deployment on consumer-grade hardware, making it suitable for both research and practical applications.

In Table 11, we present the performance of these two models on the LongBench benchmark using different KV cache compression strategies. In terms of average results, the ProtoKV method achieves the most significant performance improvement on both models. For Mistral-7B-Instruct-v0.3, ProtoKV attains an average score of 43.56, which represents a 1.6% improvement over the second-best compression method, ChunkKV (42.89). On the Phi-3.5-mini model, ProtoKV achieves an average score of 18.49, outperforming the second-best SnapKV method (17.80) by 3.9%. These results indicate that ProtoKV delivers consistent and significant performance improvements across different recent models.

| Model | Size | Method | Single-Document QA | | | Multi-Document QA | | | Summarization | | | Few-shot Learning | | | Synthetic | | Code | | Avg. |
|---|---|---|---|---|---|---|---|---|---|---|---|---|---|---|---|---|---|---|---|
| | | | NrtvQA | Qasper | MF-en | HotpotQA | 2WikiMQA | Musique | GovReport | QMSum | MultiNews | TREC | TriviaQA | SAMSum | PCount | PRe | Lcc | RB-P | |
| | | | 18409 | 3619 | 4559 | 9151 | 4887 | 11214 | 8734 | 10614 | 2113 | 5177 | 8209 | 6258 | 11141 | 9289 | 1235 | 4206 | – |
| Phi-3.5-mini -instruct | – | FullKV | 22.17 | 22.29 | 37.63 | 25.19 | 27.59 | 15.21 | 32.41 | 21.96 | 23.39 | 67.50 | 86.34 | 15.32 | 3.65 | 79.79 | 34.89 | 41.53 | 37.43 |
| | 1024 | SLM | 3.52 | 11.69 | 10.64 | 8.58 | 7.29 | 3.88 | 26.75 | 11.00 | 14.68 | 48.84 | 4.26 | 6.94 | 1.67 | 5.12 | 17.61 | 15.51 | 12.38 |
| | | H2O | 5.63 | 11.77 | 13.41 | 13.92 | 10.00 | 4.82 | 25.35 | 15.54 | 15.44 | 28.75 | 15.67 | 19.24 | 2.54 | 37.58 | 18.62 | 14.67 | 15.81 |
| | | SnapKV | 5.67 | 16.65 | 17.73 | 18.56 | 15.24 | 7.56 | 29.22 | 16.06 | 15.42 | 48.18 | 20.95 | 20.11 | 2.52 | 16.00 | 17.88 | 17.04 | 17.80 |
| | | ChunkKV | 6.18 | 16.32 | 16.18 | 16.25 | 14.60 | 7.12 | 27.49 | 17.74 | 16.45 | 48.73 | 21.25 | 19.57 | 3.21 | 15.00 | 18.99 | 16.73 | 17.61 |
| | | Pyramid | 6.00 | 16.38 | 16.30 | 16.79 | 13.81 | 7.13 | 27.40 | 17.00 | 16.18 | 49.29 | 21.86 | 19.20 | 2.83 | 15.50 | 18.16 | 16.28 | 17.51 |
| | | ProtoKV | 6.26 | 17.05 | 16.95 | 17.87 | 16.28 | 8.35 | 28.91 | 18.32 | 16.92 | 49.37 | 22.05 | 20.74 | 3.37 | 16.00 | 19.84 | 17.61 | 18.49 |
| Mistral-7B -Instruct-v0.3 | – | FullKV | 25.56 | 40.03 | 51.40 | 45.28 | 36.39 | 24.53 | 33.85 | 24.51 | 27.16 | 73.00 | 91.33 | 50.65 | 6.50 | 93.08 | 62.13 | 62.70 | 46.76 |
| | 1024 | SLM | 18.99 | 24.40 | 38.23 | 32.57 | 20.04 | 14.73 | 29.02 | 19.35 | 26.22 | 69.50 | 68.44 | 18.98 | 6.00 | 8.83 | 38.03 | 34.58 | 29.24 |
| | | H2O | 23.11 | 32.99 | 48.33 | 42.66 | 34.23 | 22.19 | 28.09 | 22.88 | 25.57 | 48.00 | 88.95 | 46.81 | 3.00 | 64.50 | 58.25 | 55.58 | 42.08 |
| | | SnapKV | 23.89 | 36.28 | 49.38 | 43.63 | 35.28 | 22.95 | 28.59 | 23.34 | 25.20 | 72.50 | 88.75 | 47.01 | 4.50 | 64.50 | 58.81 | 58.39 | 42.69 |
| | | ChunkKV | 23.51 | 36.11 | 49.49 | 44.89 | 35.49 | 23.45 | 29.32 | 22.86 | 25.47 | 72.00 | 88.76 | 47.54 | 4.50 | 64.50 | 59.84 | 58.45 | 42.89 |
| | | Pyramid | 23.43 | 36.04 | 49.67 | 44.81 | 35.67 | 23.12 | 29.38 | 22.99 | 26.13 | 72.00 | 88.91 | 47.18 | 4.50 | 64.50 | 58.98 | 58.12 | 42.84 |
| | | ProtoKV | 23.58 | 36.50 | 50.63 | 45.13 | 36.18 | 24.34 | 30.31 | 23.82 | 26.90 | 73.00 | 89.75 | 48.32 | 5.00 | 64.50 | 59.69 | 59.36 | 43.56 |

Table 11: Performance comparison on the LongBench dataset for two recent LLMs.

