# OpenReview forum: "ProtoKV: Long-context Knowledges Are Already Well-Organized Before Your Query"
_ICLR.cc/2026/Conference — ICLR 2026 Poster_

### Official Review · Reviewer_m3F1 · 2025-10-22

**Soundness:** 3
**Presentation:** 4
**Contribution:** 2
**Rating:** 6
**Confidence:** 4

**Summary:**

The paper introduces ProtoKV, a KV-cache compression method for LLMs. It first exploits a key structural insight: most tokens (“position-determined”) cluster with their neighbors, while a small set of “semantic-anchored” tokens (SAT) consistently deviate and form clusters. By splitting tokens into these two classes, ProtoKV builds separate semantic prototypes and compresses each class into semantically coherent clusters, preserving meaning while cutting memory. On LongBench it outperforms prior state-of-the-art techniques by 2.11\% accuracy under the same memory budget.

**Strengths:**

1. The paper is well written and excellently presented. Figure 1 clearly illustrates the paper's novel findings.

2. The paper finds that semantic-anchored tokens, which exhibit the local deviation property, are important for generation. It provides exhaustive experimental results to demonstrate this.

3. ProtoKV builds separate semantic prototypes and compresses each class into semantically coherent clusters. This method improves accuracy compared with existing baselines.

**Weaknesses:**

1. The rationale behind locality-sensitive hashing applied to SAT is unclear. (Question 1)

2. The time cost of ProtoKV may hinder its applicability. (Question 2)

3. There are no error bounds for the key results. Quantitative robustness indicators such as standard deviations would help validate generalizable conclusions.

**Questions:**

1. The paper claims that SATs are salient for generation, so why not select all SATs directly? The results as shown in Figure 12 illustrate that the SAT prototype number does not influence the accuracy, because ProtoKV selects all SATs, even though they are classified into different clusters. Additionally, what type of RFF-based hashing is it, locality-sensitive or random?

2. Figure 14(b) shows that the average compression time exceeds half an hour. However, full attention computation usually takes several minutes for a long context. Do the users have to wait half an hour for KV cache compression?

3. As shown in Figure 34, none of the methods match FullKV in terms of accuracy. Could ProtoKV achieves FullKV’s level of precision with a larger budget size, e.g., 1024 tokens?



Minor comments:
- Line 461, Eq. equation 8 -> Eq. 8
- The caption of Figure 34 states that bold indicates the best performance and underline the second performance, but no text in the figure is bolded or underlined.

---

> ### Author Response · Authors · 2025-11-20
>
> **Response to Question 1:**
>
> We appreciate this question about our design choices. *Section 4.1 discusses the possibility of selecting KV pairs based solely on their outlier degrees* (which would retain all SATs), this approach has a fundamental limitation: it overlooks the detailed, task-relevant information embedded in Position-Determined Tokens (PDTs). ProtoKV is designed to preserve semantic integrity by balancing the retention of high-level semantic anchors (SATs) with fine-grained contextual details from PDTs.
>
> Regarding the hashing technique, our RFF-based method is locality-sensitive, which has been proved by previous works [1]. This choice is motivated by the need to handle diverse key embedding distributions. As analyzed in Appendix E, SATs can form multiple distinct clusters in *semantically discontinuous texts*. LSH effectively addresses such multi-cluster scenarios.
>
>
> The stability shown in Figure 12 (where performance plateaus with few or even single SAT prototypes) occurs because *most LongBench sequences are semantically coherent*, making a single cluster sufficient. **The LSH mechanism is proposed based on our belief that it can offer ProtoKV with robustness for more complex, discontinuous contexts, even if this capability cannot significantly boost scores on current benchmarks like longbench. **
>
> **Response to Question 2:**
>
> We apologize for any confusion. The values in Figure 14(b) represent the **overall computation time for entire LongBench tasks**, not the time for KV cache compression only. The actual KV cache compression overhead constitutes less than 10% of this total time and is incurred only once during the pre-filling stage. Moreover, KV cache compression significantly reduces the time required for decoding, which can even shorten user's waiting time. For example, Llama3-8B with ProtoKV requires approximately 0.81 hours per LongBench task on average, while FullKV (without KV cache compression) takes about 0.84 hours. We have clarified this point in the revised manuscript.
>
> **Response to Question 3:**
>
> Yes, absolutely. We evaluated performance using **budget percentages** relative to total cache size. The results below demonstrate ProtoKV's scalability on LongBench using Llama3-8B:
>
> | Method    | 10% Budget | 20% Budget | 30% Budget |
> |-----------|------------|------------|------------|
> | *FullKV*    | 42.34      | 42.34      | 42.34      |
> | *ProtoKV*   | 41.66      | 42.27      | 42.49      |
> | *Improvement* | -1.61%     | -0.16%     | +0.37%     |
>
> Given LongBench's average input length of ~10K tokens, a 20% budget corresponds to ~2048 tokens. The data shows that *ProtoKV essentially matches FullKV performance at the 20% budget level*. We have added this part of experiment in **Appendix R in the revised PDF**.
>
> **Conlusion**
> Thank you again for these thoughtful questions. We hope our responses adequately address your concerns.
>
> [1] Locality-sensitive binary codes from shift-invariant kernels.

---

> > ### Comment · Reviewer_m3F1 · 2025-11-26
> >
> > I would thank the authors for answering Question 3. Before discussion, I would like to state the following principle: If I were a user of ProtoKV, I would not want to compromise on quality. Therefore, I would set the budget percentage to 20-30 % to avoid quality degradation.
> >
> > ## Response 1
> >
> > This response does not convince me. I insist that all SATs should be selected. ProtoKV could save the compression time by simplifying the selection method. You can persuade me by showing me an example of ProtoKV selecting only some SATs and performing better (show me the prompt, the generated answer with some SATs selected, and the generated answer with all SATs selected). It should be easy for you to find an example by examining whether all SATs are selected if such example exists.
> >
> > If you have enough time, I recommend trying this method to improve ProtoKV's efficiency.
> >
> > ## Response 2
> >
> > Recording the overall computation time is uncommon. For clarity, the authors should break it down into prefill time, decode time, and compression time. It is better to show the time distribution across the dataset and compare ProtoKV with FullKV.
> >
> > Additionally, according to my principle, if the budget percentage is set to 20-30 %, ProtoKV's generation latency may exceed that of FullKV.
> >
> > ## Response 3
> >
> > I am glad to see that ProtoKV can match FullKV's performance. However, how do prior methods perform with a budget percentage of 20-30 %? If they can also match FullKV's performance, I would prefer to use them for efficiency reasons, such as H2O.
> >
> > Taking all of the above into consideration, I will keep my score. To conclude, I appreciate the paper's findings, but I am skeptical about its applicability.

---

> ### Author Response · Authors · 2025-11-27
>
> We sincerely thank the reviewer for your appreciation of our findings. Below, we provide detailed responses to each of your concerns.
>
> ### **Response 1:**
> We apologize for the lack of clarity in our previous response. Following your suggestion, we provide an example below where selecting only a subset of SATs achieves better performance (under budget size of 20%).
>
> Prompt:
> > You are given a scientific article and a question.
> >
> > Introduction
> >
> >Semantic Role Labeling (SRL) has emerged as an important task in Natural Language Processing (NLP) due to its applicability in information extraction, question answering, and other NLP tasks. SRL is the problem of finding predicate-argument structure in a sentence, as illustrated below.
> >
> >...
> >
> >Answer the question as concisely as you can, using a single phrase or sentence if possible.
> >
> >Question: What does an individual model consist of?
>
> Ground Truth:
> >Bayesian model of garg2012unsupervised as our base monolingual model.
>
> ProtoKV:
> >The individual model consists of individual Bayesian models for each language and Bayesian model of garg2012unsupervised as base monolingual model.
>
> All SATs are preserved:
> >The individual model consists of individual Bayesian models for each language and crosslingual latent variables.
>
> We conjecture that the performance drop after **adding extra SATs** occurs because answering this question only requires information from one paragraph. Since SATs aggregate content from different paragraphs, including all of them may introduce extra noise. We suspect the phrase "crosslingual latent variables" were introduced this way.
>
> ### **Response 2:**
> As suggested, we provide a detailed time analysis below. TTFT (Time to First Token) represents the time taken to generate the first token. **TTFT of FullKV corresponds to the prefilling time**, while for ProtoKV, TTFT can be regarded as the sum of prefilling time and KV compression time. Throughput indicates the number of tokens generated per second during the decoding process.
>
> #### SETTING: input length of 12288; output length of 128; budget size of 30%
>
> | model      | TTFT (s)   | compression time (s)| throughput | decoding time (s) | total time (s) |
> |------------|-----------|-----------|------------|----------|--------- |
> | FullKV     |  1.337  | - | 34.32421   | 3.729     | 5.066   |
> | ProtoKV   |  1.385   | 0.048 | 38.93845   | 3.287       | 4.672   |
>
>
> As can be seen from the above table, even with budget size of 30%, when the output length is moderate (128 tokens), ProtoKV still achieves higher efficiency due to its greater throughput. But when the output length is short (less than 13 tokens), as you mentioned, ProtoKV's generation latency will exceed that of FullKV. **Therefore, ProtoKV is better suited for scenarios with longer output lengths.**
>
>
> ### **Response 3:**
> Following your concern, we also compared ProtoKV with H2O and SnapKV with larger KV cache budgets:
>
> | Method   | 10% Budget         | 20% Budget         | 30% Budget         |
> |----------|--------------------|--------------------|--------------------|
> | FullKV   | 42.34              | 42.34              | 42.34              |
> | ProtoKV  | 41.66 (–1.61%)     | **42.27 (–0.16%)** | **42.49 (+0.37%)** |
> | H2O      | 38.09 (–10.04%)    | 38.66 (–8.69%)     | 40.51 (–4.32%)     |
> | SnapKV   | 40.83 (–3.57%)     | 41.92 (–0.99%)     | **42.26 (–0.19%)** |
>
> As shown, H2O still lags behind FullKV even at 30% budget, while SnapKV requires more than 30% KV budget to match FullKV’s performance. This indicates that our ProtoKV remains effective under larger KV budget size.
>
> ### **Conclusion:**
> We are grateful for your insightful comments, which guided us to refine our analysis especially under scenarios with sufficient KV budget. We hope these results can alleviate your concerns. Thank you again for your valuable time and insights.

---

### Official Review · Reviewer_KnCx · 2025-10-28

**Soundness:** 3
**Presentation:** 3
**Contribution:** 3
**Rating:** 6
**Confidence:** 3

**Summary:**

The paper introduces ProtoKV, a novel framework designed to enhance the efficiency of Key-Value (KV) cache retention in large language models (LLMs) when processing long text sequences. The authors identify two categories of tokens within the key embedding space: Position-Determined Tokens (PDTs), which maintain strong similarity with their contextual neighbors, and Semantic-Anchored Tokens (SATs), which exhibit local deviation and form clusters. By leveraging the unique properties of these two token types, ProtoKV constructs semantic prototypes that improve KV cache compression while preserving semantic integrity. Experimental results demonstrate that ProtoKV outperforms existing state-of-the-art methods by achieving an average accuracy improvement of 2.11% on the LongBench benchmark, showcasing its effectiveness in maintaining high retrieval accuracy with minimal KV cache retention.

**Strengths:**

- Good Presentation. The presentation of this paper is very clear, the structure is reasonable, and the presentation of the figures and tables is also very precise.
- Reasonable Idea:  The paper presents a novel perspective on token categorization in LLMs, particularly the identification and utilization of SATs as semantic anchors, which is a significant contribution to the field.
- Good Performance: The authors provide comprehensive experiments across various benchmarks, clearly demonstrating the advantages of ProtoKV over existing methods in terms of accuracy and efficiency.
- Insightful Analyses.

**Weaknesses:**

- Complexity of Implementation: The proposed method may introduce additional complexity in the implementation of LLMs, which could be a barrier for adoption in certain applications or by practitioners with limited resources.
- Limited Comparison with Other Methods: While the paper provides lots of experiments to evaluate ProtoKV, a more extensive analysis involving **more recent SOTA approaches** could strengthen the argument for its superiority.

**Questions:**

Please refer to the weakness part.

---

> ### Author Response · Authors · 2025-11-20
>
> **Response to Question 1:**
>
> Figure 14 demonstrates that Llama3-8B with ProtoKV requires approximately 0.81 hours per LongBench task on average, while our additional experiments show that FullKV (without KV cache compression) takes about 0.84 hours. This indicates that our method not only reduces memory overhead by compressing the KV cache but also decreases inference time. Although ProtoKV introduces additional compression overhead during the prefilling stage, the computational cost during generation is significantly reduced due to the compressed KV cache. Consequently, the overall inference time is reduced. Therefore, ProtoKV is particularly advantageous for applications, especially in resource-constrained scenarios.
>
>
> **Response to Question 2:**
>
> Among the baselines we compared, ClusterKV and SqueezeAtt are works from late last year, while ChunkKV and SentenceKV are from this year. We have reviewed several recent works, and notable approaches such as SepLLM [1] and the latest DeepSeek-V3.2 [2] for long-context handling involve architectural modifications or post-training optimizations. In contrast, our method does not require adjustments to the LLM parameters. Methods like SubKV [3] and A2ATS [4] employ quantization techniques, while SpindleKV [5] and ZigZagKV [6] fall into the category of KV Cache Budget Allocation. Both types of methods are orthogonal to our eviction strategy—meaning they can be combined with our approach to achieve better compression. We have conducted relevant experiments in Section 5.3 to validate the compatibility of our method.
>
> **Conlusion**
> Thank you again for these thoughtful questions. We hope our responses adequately address your concerns.
>
> [1] SepLLM: Accelerate Large Language Models by Compressing One Segment into One Separator
>
> [2] DeepSeek-V3 Technical Report
>
> [3] Subkv: Quantizing Long Context KV Cache for Sub-Billion Parameter Language Models on Edge Devices
>
> [4] A2ATS: Retrieval-Based KV Cache Reduction via Windowed Rotary Position Embedding and Query-Aware Vector Quantization
>
> [5] SpindleKV: A Novel KV Cache Reduction Method Balancing Both Shallow and Deep Layers
>
> [6] ZigZagKV: Dynamic KV Cache Compression for Long-context Modeling based on Layer Uncertainty

---

> ### Comment · Reviewer_KnCx · 2025-11-25
> **final rating**
>
> Thanks for your clarification, and the rebuttal addresses most of my concerns. I tend to accept this paper and keep my inital rating (6).

---

> > ### Author Response · Authors · 2025-11-25
> >
> > Thank you for your recognition and support. We are delighted to hear that our rebuttal addressed your concerns. Wishing you all the best in your work and life!

---

### Official Review · Reviewer_Y76g · 2025-10-31

**Soundness:** 2
**Presentation:** 2
**Contribution:** 2
**Rating:** 4
**Confidence:** 2

**Summary:**

The paper proposes ProtoKV, a semantic-aware KV cache compression framework for large language models (LLMs) that mitigates long-context inference costs. It introduces two token categories—Semantic-Anchored Tokens (SATs) and Position-Determined Tokens (PDTs)—and constructs hybrid semantic prototypes for each, guiding KV retention through cluster-based attention relevance. The method preserves semantic integrity while maintaining computational efficiency, outperforming baselines such as SnapKV, H2O, and ChunkKV by up to 2.11% on LongBench and achieving 97.3% retrieval accuracy in Needle-in-a-Haystack tests.

**Strengths:**

•	The identification of SATs as clustering outliers in the key embedding space (Fig. 4–6) provides a new lens for understanding token semantics in LLMs. This insight grounds ProtoKV’s prototype-based compression design and distinguishes it from previous attention- or position-driven methods.
•	The framework (Sec. 4.2–4.3) integrates Random Fourier Feature hashing (Eq. 7–8) and prototype-guided selection (Eq. 10–11), avoiding costly iterative clustering (Fig. 8). Pseudocode and reproducibility details are given (Appendix J), enhancing transparency.
•	ProtoKV is compared with multiple baselines across three architectures (LLaMA-2, LLaMA-3, Mistral) and two benchmarks (LongBench, Ruler), showing robustness under varying KV budgets (64–512) and across tasks (Fig. 9, 10, 12–14). The ablation studies further isolate the roles of prototype number and SAT count.

**Weaknesses:**

•	While the “local deviation property” (Eq. 4–6) is empirically supported, the causal explanation (Sec. 3.2) remains qualitative. There is no analytical or statistical validation that SATs correspond to meaningful semantic units across layers or models.
•	Key hyperparameters such as neighborhood window $\kappa$, prototype number, and threshold $\beta$ are only briefly tuned (Fig. 12–13) without robustness metrics or cross-dataset variance, limiting confidence in generalizability.
•	Although computational cost is compared (Fig. 14), there is no wall-clock latency or memory breakdown versus model size (e.g., >8 B models), and no statistical significance tests for accuracy gains (Table 2–4).
•	The contribution of each stage (SAT detection, LSH clustering, observation window) is only partially evaluated; removing or modifying these modules’ effects is not explicitly quantified.

**Questions:**

1.	Could the authors provide layer-wise or head-wise distributions of SATs to clarify whether semantic anchoring is model-general or architecture-specific?
2.	How does ProtoKV behave under streaming or dynamic decoding, where prefilling-only assumptions may not hold?
3.	Can the authors include significance analysis or confidence intervals to verify the reported 2.11 % average improvement?

---

> ### Author Response · Authors · 2025-11-20
>
> **Response to Question 1:**
>
> The phenomenon of semantic anchoring is model-general and persists across various large model architectures. To validate this, we conducted additional experiments detailed in **Appendix N in the revised PDF**. We evaluated representative models from three key attention architectures: 1) the original multi-head attention (LlaMa2-7B), 2) Grouped-Query Attention (Qwen3-4B), and 3) Multi-Query Attention (StarCoder). Additionally, we tested the open-source Mixture-of-Experts model GPTOSS-20B, which implements Flash Attention-with-Sink. Results in Appendix N demonstrate that: 1) Semantic anchoring occurs consistently across all tested architectures, including the MoE model, evidenced by the presence of tokens with high outlier degrees during long-context processing. 2) We also provide layer-wise distributions of Semantic Anchor Tokens (SATs), visualized by counting outlier keys per KV head within each layer and plotting their distribution via boxplots (thresholds are set as 1.5, 1.75, 2 respectively). The visualizations show that most layers and heads contain outlier key vectors (SATs), which fully consist with our main conclusions.
>
> **Response to Question 2:**
>
> For dynamic decoding setting, we introduce a segment-wise KV cache compression strategy that employs a buffer mechanism to process text in segments, and performs compression after each segment while retaining observation tokens. The detailed workflow is described in **Appendix O in the revised PDF**. We report results for math reasoning datasets of MATH and AIME with LlaMa3-8B as follows:
>
> | Benchmark | Method   | KV Size 1024 | KV Size 2048 |
> |:-----------|:----------|:----------------|:----------------|
> |  MATH     | H2O      | 71.64          | 78.15          |
> |           | SnapKV   | 74.43          | 80.50          |
> |           | ChunkKV  | 76.69          | 80.42          |
> |           | ProtoKV  | 79.67          | 81.39          |
> |  AIME     | H2O      | 19.51          | 29.87          |
> |           | SnapKV   | 17.73          | 32.76          |
> |           | ChunkKV  | 18.32          | 34.98          |
> |           | ProtoKV  | 21.05          | 38.26          |
>
> Our approach consistently outperforms existing techniques. For example, with a 1024 KV budget, ProtoKV surpasses H2O by 8.03%, SnapKV by 5.24%, and ChunkKV by 2.98%. The advantage persists with a 2048 budget. On the more challenging AIME benchmark, improvements are even more significant: at a 1024 budget, ProtoKV outperforms H2O, SnapKV, and ChunkKV by 7.89%, 18.73%, and 14.90%, respectively. This indicates that ProtoKV remains effective under dynamic decoding scenarios.

---

> > ### Author Response · Authors · 2025-11-20
> >
> > **Response to Question 3:**
> >
> > Thank you for the suggestion. To rigorously validate that the performance improvements of our method are solid and not merely attributable to the inherent stochasticity of LLMs, we configure the model to perform sampling during the decoding phase, and set the generation temperature to 0.6 and sampling 5 different model outputs. We report the results on Longbench for LlaMa3-8B, where the variance can reflect confidence intervals.
> > ##### Size: 128
> >
> > | Model    | SDQA           | MDQA           | SUM            | FSL            | Synthetic      | Code           |
> > |----------|----------------|----------------|----------------|----------------|----------------|----------------|
> > | SLM      | 15.38±2.97     | 24.36±3.59     | 19.47±1.19     | 48.61±5.12     | 37.03±0.32     | 47.31±3.16     |
> > | H2O      | 20.84±1.56     | 27.01±2.78     | 18.12±2.27     | 53.57±3.32     | 37.45±0.42     | 52.81±1.03     |
> > | SnapKV   | 23.19±0.59     | 29.03±1.03     | 20.03±0.62     | 60.36±0.98     | 37.03±0.11     | 54.12±0.87     |
> > | ChunkKV  | 23.04±0.68     | 29.48±0.87     | 20.23±0.29     | 61.20±1.51     | 36.95±0.09     | 53.73±1.13     |
> > | Pyramid  | 22.37±0.57     | 29.12±0.99     | 20.31±0.44     | 60.55±1.13     | 36.82±0.09     | 54.54±0.92     |
> > | ProtoKV  | 23.69±0.32     | 29.67±0.71     | 20.92±0.31     | 62.17±0.67     | 37.20±0.06     | 56.61±0.57     |
> >
> > ##### Size: 1024
> >
> > | Model    | SDQA           | MDQA           | SUM            | FSL            | Synthetic      | Code           |
> > |----------|----------------|----------------|----------------|----------------|----------------|----------------|
> > | SLM      | 22.69±3.15     | 28.99±4.32     | 19.56±2.39     | 52.50±2.98     | 37.39±0.06     | 50.68±2.29     |
> > | H2O      | 25.70±1.94     | 30.81±1.79     | 20.93±1.26     | 59.18±3.05     | 37.62±0.25     | 54.89±1.67     |
> > | SnapKV   | 27.11±0.79     | 32.19±0.93     | 23.04±0.31     | 67.06±0.82     | 36.95±0.12     | 56.30±1.61     |
> > | ChunkKV  | 27.83±0.87     | 32.08±0.84     | 22.21±0.26     | 66.54±0.64     | 36.82±0.31     | 57.62±0.83     |
> > | Pyramid  | 26.90±0.65     | 31.47±0.62     | 22.62±0.28     | 66.32±0.91     | 36.75±0.68     | 55.87±2.01     |
> > | ProtoKV  | 28.29±0.44     | 33.56±0.65     | 23.47±0.10     | 67.84±0.41     | 37.87±0.02     | 58.59±0.69     |
> >
> > Results are visualized in **Appendix Q in the revised PDF**. Our proposed ProtoKV exhibits low variance, while other baselines show relatively higher variance. Crucially, the lower performance bound of ProtoKV even exceeds that of all baselines for tasks like summarization and code generation, demonstrating its robustness.
> >
> > **Conclusion:**
> > Thank you again for your valuable comments. We hope our revised manuscript and the new experiments can address your concerns. **We have gone to great lengths to complete these analyses within the rebuttal period, and we appreciate your understanding regarding any remaining limitations.** Your feedback has been immensely helpful, and we are hopeful for your positive assessment.

---

> ### Author Response · Authors · 2025-11-23
>
> I hope this message finds you well. I wanted to kindly follow up regarding my rebuttal. I truly appreciate the time and effort you’ve dedicated to reviewing my work, and I was hoping you might have a chance to review my responses and provide any additional feedback or updates to your evaluation at your earliest convenience. Thank you again for your valuable time and insights.

---

### Official Review · Reviewer_mS4R · 2025-11-04

**Soundness:** 2
**Presentation:** 2
**Contribution:** 2
**Rating:** 4
**Confidence:** 3

**Summary:**

The authors discover that while most tokens demonstrate high similarity (in key space) with their contextual neighbors (position-determined tokens, PDTs), a subset of tokens (dubbed "semantic-anchored tokens", SATs) deviate from this property while accumulating a significant amount of attention. The authors construct lsh-based prototypes for SATs and chunk-based protoypes for PDTs as compression units. Clusters are ranked according to an importance metric, tokens are assigned to these clusters, and tokens from the top-ranked clusters are retained until the budget is met. This approach (ProtoKV) outcompetes baselines by >2% on LongBench and outcompetes other baselines at a lower budget and ties others at higher budgets.

**Strengths:**

- This appears to be the first work to discover SATs. The experiment distinguishing them from sinks validates this unique token type.

- The clustering approach is principled according to the token types.

- ProtoKV defeats several popular strategies across varying model families on LongBench.

- The success of the method at a very small budget (64 tokens) is attractive for severely resource-constrained systems.

**Weaknesses:**

- LongBench and RULER are known to stack lots of noisy context around sparsely distributed signals, thus possibly rendering the appearance of SATs as unique to these types of benchmarks. The appearance of this token type does not appear to be explored over a greater variety of long-context tasks.

- Besides Llama-3-8B Instruct, only older models are tested. The authors should consider evaluating their approach on newer Qwen, Phi models, and/or the latest Mistral-7B.

- The performance gain on RULER is quite minimal. While the average improvement on LongBench is +2%, the individual numbers on LongBench in Figure 34 are far less significant, where ProtoKV either incrementally wins or even loses against other baselines on a variety of tasks. This makes it difficult to determine whether ProtoKV is truly a worthwhile compression strategy.

**Questions:**

- See weaknesses.
 - How does the method perform on RULER 16K or 32K?
 - Is H2O truly **that** bad on RULER (Table 2)? This doesn't seem to concur with other literature.
 - How does this approach fundamentally differ from the Reformer, which also chunks and groups tokens according to LSH buckets?

---

> ### Author Response · Authors · 2025-11-20
>
> **Response to Question 1:**
>
> *Response to Weakness 1:*
>
> Thank you for the constructive suggestion. To further verify the prevalence of SATs, we performed additional key distribution analysis on the AIME and HMMT datasets—both are challenging math problems and contain rigorous, logically sound solutions with **minimal noise**. We formatted each problem as *"Question: [...], Solution: [...]"* for model prefilling and analyzed the resulting key vector distributions. Visualization results in **Appendix N in revised PDF** show that SATs (reflected by high outlier degrees in key vectors) remain prevalent across various model architectures (including MHA, MQA, GQA), supporting the generalizability of our findings across diverse long-context tasks.
>
> *Response to Weakness 2:*
>
> We have conducted additional experiments on LongBench using the more recent Phi-3.5-Mini-Instruct and Mistral-7B-Instruct-v0.3 models with budget size of 1024. Average results are reported below:
> | Model | SLM | H2O | SnapKV | ChunkKV | Pyramid | ProtoKV |
> |:---|:---:|:---:|:---:|:---:|:---:|:---:|
> | **Phi-3.5-mini-instruct** | **12.38** | **15.81** | **17.80** | **17.61** | **17.51** | **18.49** |
> | **Mistral-7B-Instruct-v0.3** | **29.24** | **42.08** | **42.69** | **42.89** | **42.84** | **43.56** |
>
> For Mistral-7B-Instruct-v0.3, ProtoKV achieves an average score of 43.56, a 1.6% improvement over the second-best method, ChunkKV (42.89). For Phi-3.5-Mini, ProtoKV attains an average score of 18.49, outperforming the second-best SnapKV (17.80) by 3.9%. We have added the detailed results for every single task in **Appendix S in revised PDF**.
>
> *Response to Weakness 3:*
>
> We acknowledge the reviewer's question regarding the 2% improvement. To contextualize this result, we estimate an empirical upper bound for KV cache eviction by introducing an Oracle KV strategy: we append the correct answer after the context, retain the top key-value pairs by Inference-stage Cumulative Attention with Eq. (1) in our paper, and regenerate the output. Experiments were conducted on LLaMA3-8B, with results summarized below:
>
> | Budget Size | 128    | 256    | 512    | 1024   |
> |-------------|--------|--------|--------|--------|
> | ProtoKV     | 37.52  | 40.13  | 40.76  | 41.58  |
> | Oracle KV   | 38.41  | 40.96  | 41.05  | 42.26  |
> | Gap  | 2.37%  | 2.07%  | 0.71%  | 1.63%  |
>
> As shown, the performance gap between Oracle KV and our method is minimal (~1.7% across different budgets). This suggests that our ProtoKV is already operating near the theoretical upper bound, and therefore the 2% improvement over baseline methods represents a meaningful gain within practical constraints.
>
>
> **Response to Question 2:**
>
> Results for the LlaMa3-8B model on RULER 16K and 32K with budget size of 512 are provided below.
>
> | Method | RULER-16K | RULER-32K |
> |---|---|---|
> | **H2O** | **27.4** | **24.6** |
> | **SnapKV** | **64.0** | **53.4** |
> | **ChunkKV** | **65.9** | **54.7** |
> | **PyramidKV** | **64.2** | **53.9** |
> | **ProtoKV** | **68.3** | **57.2** |
>
> ProtoKV maintains consistent and non-trivial improvements: 6.7–7.1% over SnapKV, 3.6–4.6% over ChunkKV, and 6.1–6.4% over PyramidKV. These results are significantly better than those on the 4K RULER dataset tested in the main paper. We have added it in **Appendix P in revised PDF**. Thank you for your suggestion and guidance!
>
>
> **Response to Question 3:**
>
> H2O indeed underperforms on the RULER dataset. Previous study [1] has also supported this observation. A possible reason is that RULER requires precise localization of answers in the text according to the question, while H2O employs a question-agnostic KV cache retention strategy.
>
> **Response to Question 4:**
>
> Reformer involves architectural modifications to enhance long-context handling after pretraining. In contrast, our approach focuses on KV cache compression for **fixed-architecture, fixed-parameter LLMs**. This is the key difference. Additionally, our proposed ProtoKV applies LSH only to a small set of semantic anchor tokens, whereas Reformer applies it to all tokens. As shown in Figure 11, applying LSH to all tokens in ProtoKV leads to poor performance, as our method is not adapted during pretraining.
>
> **Conclusion:**
> Thank you again for your valuable comments. We hope our revised manuscript and the new experiments can address your concerns. **We have gone to great lengths to complete these analyses within the rebuttal period, and we appreciate your understanding regarding any remaining limitations.** Your feedback has been immensely helpful, and we are hopeful for your positive assessment.
>
> [1] A2ATS: Retrieval-Based KV Cache Reduction via Windowed Rotary Position Embedding and Query-Aware Vector Quantization

---

> ### Author Response · Authors · 2025-11-23
>
> I hope this message finds you well. I wanted to kindly follow up regarding my rebuttal. I truly appreciate the time and effort you’ve dedicated to reviewing my work, and I was hoping you might have a chance to review my responses and provide any additional feedback or updates to your evaluation at your earliest convenience. Thank you again for your valuable time and insights.

---

> > ### Comment · Reviewer_mS4R · 2025-11-24
> > **Thanks for the Rebuttal**
> >
> > I thank the authors for their response and addressing many of my concerns through further experiments.
> >
> > I have one further comment: the "empirical upper bound" set by OracleKV is not necessarily the best possible performance achievable for a compression strategy -- this inherently assumes that retaining the highest accumulated-attention tokens will necessarily result in the highest performance, but this is still very much an open question since an optimal token importance heuristic is still not known. What you have actually determined is that your method is closely imitates a heavy-hitter strategy that has knowledge of the correct answer. Additionally, this may be influenced by the structure and content of the correct answer.
> >
> > In any case, in light of this rebuttal I am increasing my score. Thanks!

---

> > > ### Author Response · Authors · 2025-11-25
> > >
> > > Thank you for your thoughtful follow-up. We sincerely appreciate your recognition and the time you have dedicated to reviewing our work.
> > >
> > > We fully agree that the question of what constitutes an optimal oracle strategy remains open and nuanced, which may deserve a separate research effort. Our implementation of OracleKV was intended as a heuristic approach, and we acknowledge that given the complexity and the black-box nature of LLMs, the relationship between such heuristics and final performance is not fully settled.
> > >
> > > Thank you again for your valuable feedback and encouragement throughout the review process. Wishing you all the best in your work and life!

---

### Comment · Reviewer_Y76g · 2025-11-26

Thanks to the authors for their efforts and responses. I believe most of my concerns have been addressed and will raise the score.

---

> ### Author Response · Authors · 2025-11-26
>
> Thank you for your recognition and support. We are delighted to hear that our rebuttal addressed your concerns. Wishing you all the best in your work and life!

---

### Author Response · Authors · 2025-11-30
**Summary of Our Work and Rebuttal Discussions**

Dear Area Chair,

Thank you for your valuable time. To provide a clearer understanding of our work and the discussions during the rebuttal phase, we have summarized the key points below:

This work investigates the distribution of key vectors in LLMs when processing long context inputs. We for the first time discovered the phenomenon of **local heterogeneity**: most tokens (“position-determined”) show similar key vectors with their neighbors, while a small set of **“semantic-anchored” tokens (SAT)** consistently deviate and form clusters. (Section 3). Based on this finding, by splitting tokens into these two classes, we proposed the **ProtoKV** algorithm for KV cache compression (Section 4), which significantly enhances LLM's performace in handling long contexts (Section 5).

We were pleased to see that the reviewers highlighted many of the strengths of the paper, including:
- **Novel discovery of SATs** [mS4R, Y76g, KnCx, m3F1]
- **Effectiveness of ProtoKV** [mS4R, Y76g, KnCx, m3F1]
- **Comprehensive Evaluation and Analysis** [KnCx, Y76g, m3F1]
- **Clear presentation** [KnCx, m3F1]

Reviewers also raised questions mainly regarding the **generality of the phenomenon** [mS4R, Y76g] and **lack of certain experiments** [mS4R, Y76g, KnCx, m3F1]. In response to their concerns, we have included the following additional experiments **in the revised PDF**:

- **Appendix N** validates the prevalence of the local heterogeneity phenomenon across a wider variety of model architectures (e.g., GPT-OSS) and datasets (e.g., AIME).
- **Appendix O** verifies that our ProtoKV method is equally effective in dynamic decoding scenarios.
- **Appendix P** provides the experimental results on RULER 16K/32K datasets.
- **Appendix Q** presents a confidence interval analysis with temperature sampling, indicating our enhanced robustness and more reliable model outputs compared to existing algorithms.
- **Appendix R** shows the performance of ProtoKV under larger KV cache budgets.
- **Appendix S** presents the performance of ProtoKV when adapted to other advanced LLMs as suggested.

Our responses received positive feedback from the reviewers. The reviewers' ratings were updated as follows after reviewing our rebuttal: Reviewer mS4R (**4 → 6**), Reviewer Y76g (**4 → 6**), and Reviewers KnCx and m3F1 **maintained their positive ratings of 6**. This resulted in the overall rating improving from **(4,4,6,6)** to **(6,6,6,6)**.

In response to Reviewer m3F1's further questions, we have provided a case study and a detailed breakdown analysis of inference time. However, due to the closure of the discussion window, we were unable to engage in further communication.

We thank the reviewers for their valuable suggestions, which have helped us improve our work. We also appreciate their positive feedback on our research and the discussion. We would also be most grateful if the paper could earn your recognition.

Thank you once again for your valuable time.

Sincerely,

The Authors

---

### Meta-Review · Area_Chair_Jajc · 2025-12-12

**Summary:**

(*Disclaimer: given the peculiar review process, some of my choices and reasonings below will be highly subjective, as I tried to imagine how a reviewer would have reacted to a specific response. I understand that any negative choice will be perceived as unfair by the authors, and I apologize in advance for that.*)

(*Second disclaimer: the authors and some reviewers explicitly mention some changes in scores that occurred during the rebuttal. As these were reverted due to the possibility of collusion in light of the security incident, I will tend to disregard this information.*)

The paper proposes a KV cache compression method built on the observation that tokens can belong to two different categories, relying either on contextual or semantic cues. The authors build different kind of prototypes for the two classes, showing the method outperforms state-of-the-art KV cache compression methods on multiple benchmarks.

The initial reviews were all clustered around a borderline acceptance (`m3F1`, `KnCx`) or borderline rejection (`Y76g`, `mS4R`), making this a very difficult paper. The authors provided a very significant rebuttal with many additional experiments, that answered most of the questions raised by the reviewers. However, the rebuttal did not address weaknesses that were not explicitly framed as questions (e.g., the request of error bounds from `m3F1`). I will discuss some of these concerns below.

All reviewers agree that the discovery of semantic tokens is novel, distinct from attention sinks, and relevant to the field, although (as with similar observations in the literature) it is not supported by any theoretical framework (`Y76g`). The main questions concerned the lack of certain datasets and baselines, the potentially costly algorithm, and the possibly non-significant improvements. As I argue below, I believe that most of these comments were addressed and that all reviewers would have eventually clustered around a borderline acceptance, with the potential for one or two reviewers to recommend a full acceptance (8).

**Reviewer Concerns:**

**Lack of datasets and models** (`mS4R`, `Y76g`, `KnCx`): this was the main set of concerns from the majority of reviewers. The authors added several experiments across multiple appendices, and all reviewers were satisfied by these changes. I believe these points were fully addressed.

**Significance of the results** (`mS4R`, `Y76g`): for `mS4R`, the authors provided an oracle, showing that the results were very close to the "optimum" as defined by this oracle. The reviewer was partly convinced but the discussion did not progress further. For reviewer `Y76g`, the authors provided a standard deviation across multiple generations from the same models. Overall, the authors added many experiments strengthening their case.

**Lack of theoretical motivation for SATs** (`Y76g`): not addressed during the rebuttal. However, this is a common concern for many similar papers, thus I will not consider it for my evaluation.

**Hyperparameter robustness** (`Y76g`): not addressed during the rebuttal (not sure why).

**Complexity / wall-clock time** (`Y76g`, `KnCx`, `m3F1`): mentioned by `Y76g` but not addressed; however, similar concerns were discussed with `KnCx` and `m3F1`). The answer seems convincing and this issue was mostly addressed.

**Additional scenarios** (`Y76g`, `m3F1`): `Y76g` asked for experiments on a streaming decoding setting, which were provided. `m3F1` asked for experiments where all SATs were kept and with larger KV cache budgets. All points were addressed satisfingly.

**Reviewer Scores:**

`m3F1`: the reviewer was not convinced by the initial answer to their three questions. The additional comments from the authors seem convincing to me. I believe an increase in score (e.g., 8) was possible.

`KnCx`: this was a very shallow review with limited discussion during the rebuttal.

`Y76g`: while some comments were not discussed (see above), many other questions were addressed. The reviewer also mentioned that the responses were satisfactory. Given concerns for the lack of theoretical grounding, a score of 6 is the most probable outcome.

`mS4R`: two out of three concerns were addressed. Also here, a score of 6 is the most probable outcome.

---

### Decision · Program_Chairs · 2026-01-26

Accept (Poster)